# Discovering Universal Geometry in Embeddings with ICA

**Hiroaki Yamagiwa**[* 1]     **Momose Oyama**[* 1,2]     **Hidetoshi Shimodaira**[1,2]

[1]Kyoto University     [2]RIKEN

hiroaki.yamagiwa@sys.i.kyoto-u.ac.jp,
oyama.momose@sys.i.kyoto-u.ac.jp, shimo@i.kyoto-u.ac.jp

## Abstract

This study utilizes Independent Component Analysis (ICA) to unveil a consistent semantic structure within embeddings of words or images. Our approach extracts independent semantic components from the embeddings of a pre-trained model by leveraging anisotropic information that remains after the whitening process in Principal Component Analysis (PCA). We demonstrate that each embedding can be expressed as a composition of a few intrinsic interpretable axes and that these semantic axes remain consistent across different languages, algorithms, and modalities. The discovery of a universal semantic structure in the geometric patterns of embeddings enhances our understanding of the representations in embeddings.

## 1   Introduction

Embeddings play a fundamental role in representing meaning. However, there are still many aspects of embeddings that are not fully understood. For instance, issues such as the dimensionality of embeddings, their interpretability, and the universal properties shared by embeddings trained with different algorithms or in different modalities pose important challenges in practical applications.

Discussions and research have explored the topics of low-dimensionality and interpretability of embeddings (Goldberg, 2017). Proposals have been made for learning and post-processing methods that incorporate constraints, aiming to achieve sparse embeddings or acquire semantic axes. Additionally, research has focused on aligning embeddings trained in different languages through various transformations. However, in contrast to this ongoing trend, our specific focus lies on the intrinsic independence present within embeddings.

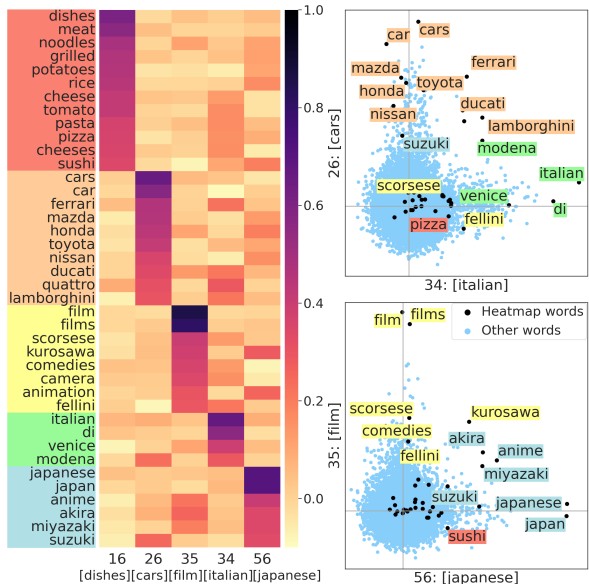

Figure 1:   (Left) Heatmap of normalized ICA-transformed word embeddings shown for a selected set of five axes out of 300 dimensions. Each axis has its own meaning, and the meaning of a word is represented as a combination of a few axes. For example, *ferrari = [cars] + [italian]* and *kurosawa = [film] + [japanese]*. (Right) Scatterplots of normalized ICA-transformed word embeddings for the (*[italian]*, *[cars]*) axes and (*[japanese]*, *[film]*) axes. The word embeddings in the heatmap were plotted as black dots. The words are highlighted with colors corresponding to their respective axes. For more details, refer to Section 3 and Appendix B.

In this research, we post-process embeddings using Independent Component Analysis (ICA), providing a new perspective on these issues (Hyvärinen and Oja, 2000). There are limited studies that have applied ICA to a set of word embeddings, with only a few exceptions (Lev et al., 2015; Albahli et al., 2022; Musil and Mareček, 2022). There has also been a study that applied ICA to word-context matrices instead of distributed representations (Honkela et al., 2010). Although it has received less attention in the past, using ICA al-

---

*The first two authors contributed equally to this work.
Our code and data are available at https://github.com/shimo-lab/Universal-Geometry-with-ICA.

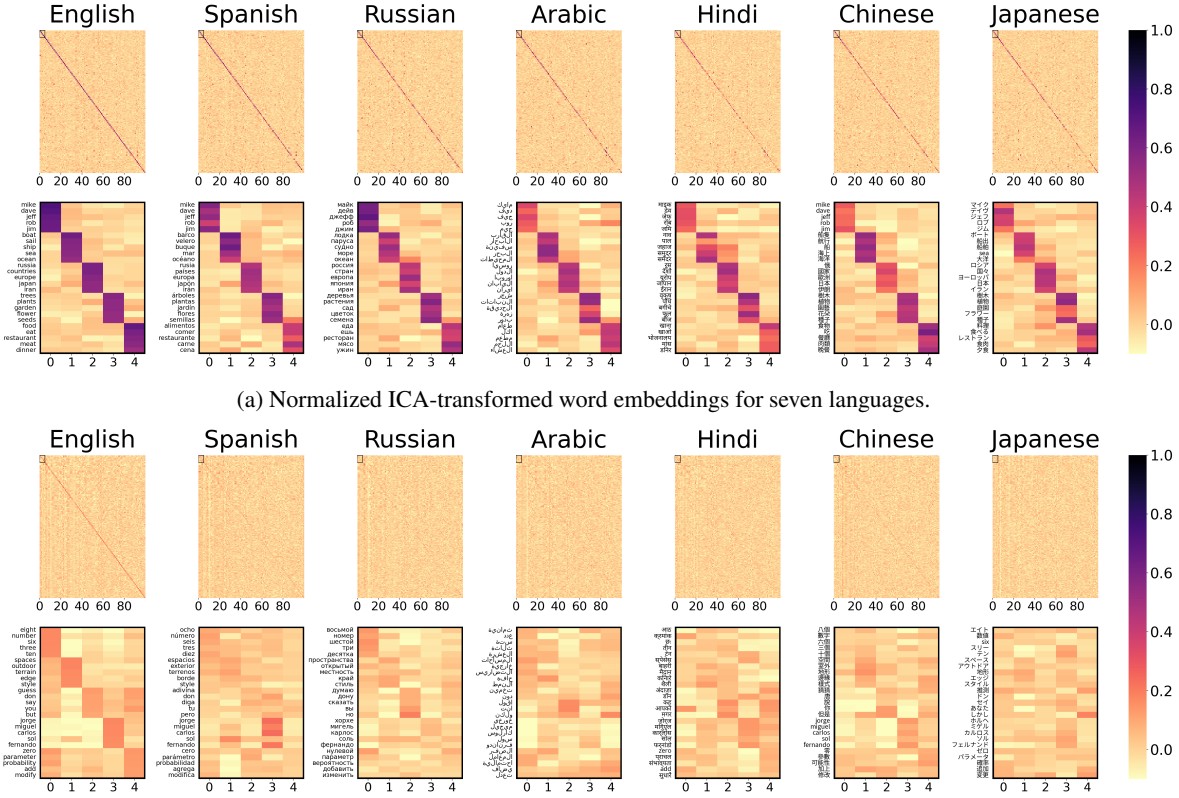

(a) Normalized ICA-transformed word embeddings for seven languages.

(b) Normalized PCA-transformed word embeddings for seven languages.

Figure 2: The rows represent fastText embeddings of words for seven languages transformed by (a) ICA and (b) PCA. Each embedding is normalized to have a norm of 1 for visualization purposes. The components from the first 100 axes of 500 embeddings are displayed in the upper panels, and the first five axes are magnified in the lower panels. For each axis of English embeddings, the top 5 words were chosen based on their component values, and their translated words were used for other languages. Correlation coefficients between transformed axes were utilized to establish axis correspondences and carry out axis permutations; English served as the reference language to align with the other languages when matching the axes, and the aligned axes were rearranged in descending order according to the average correlation coefficient, as detailed in Section 4 and Appendix C.

lows us to extract independent semantic components from a set of word embeddings. By leveraging these components as the embedding axes, we anticipate that each word can be represented as a composition of intrinsic (inherent in the original embeddings) and interpretable (sparse and consistent) axes. Our experiment suggests that the number of dimensions needed to represent each word is considerably less than the actual dimensions of the embeddings, enhancing the interpretability.

Fig. 1 shows an example of independent semantic components that are extracted by ICA. Each axis has its own meaning, and a word is represented as a combination of a few axes. Furthermore, Figs. 2a and 3a show that the semantic axes found by ICA are almost common across different languages when we applied ICA individually to the embeddings of each language. This result is not limited to language differences but also applies

when the embedding algorithms or modalities (i.e., word or image) are different.

Principal Component Analysis (PCA) has traditionally been used to identify significant axes in terms of variance, but it falls short in comparison to ICA; the patterns are less clear for PCA in Figs. 2b and 3b. Embeddings are known to be anisotropic (Ethayarajh, 2019), and their isotropy can be greatly improved by post-processes such as centering the mean, removing the top principal components (Mu and Viswanath, 2018), standardization (Timkey and van Schijndel, 2021), or whitening (Su et al., 2021), which can also lead to improved performance in downstream tasks. While the whitening obtained by PCA provides isotropic embeddings regarding the mean and covariance of components, notably, ICA has succeeded in discovering distinctive axes by focusing on the anisotropic information left in the third and higher-order mo-

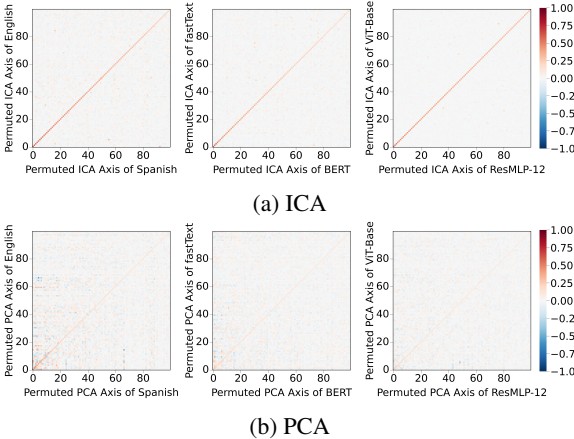

(a) ICA

(b) PCA

Figure 3: Cross-correlation coefficients for ICA and PCA-transformed embeddings. The clear diagonal lines for the ICA-transformed embeddings indicate a good alignment. (Left) English fastText and Spanish fastText shown in Fig. 2. (Middle) fastText and BERT shown in Fig. 7. (Right) Image models of ViT-Base and ResMLP-12 shown in Fig. 8.

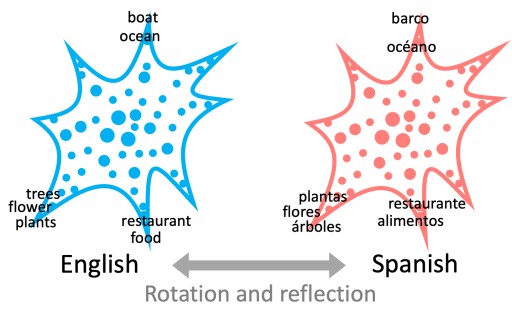

Figure 4: Illustration of the spiky shape of the word embedding distributions in high-dimensional space. This spiky shape is actually observed in the scatterplots of the normalized ICA-transformed embeddings for seven languages, as shown in Fig. 15 in Appendix C.

ments of whitened embeddings.

## 2 Background

### 2.1 Interpretability in word embeddings

The interpretability of individual dimensions in word embedding is challenging and has been the subject of various studies.

A variety of studies have adopted explicit constraints during re-training or post-processing of embeddings to improve sparsity and interpretability. This includes the design of loss functions (Arora et al., 2018), the introduction of constraint conditions by sparse overcomplete vectors (Faruqui et al., 2015), and re-training using $k$-sparse autoencoders (Makhzani and Frey, 2013; Subramanian et al., 2018), sparse coding (Murphy al.,

2012; Luo et al., 2015), or $\ell_1$-regularization (Sun et al., 2016). Additionally, the sense polar approach designs objective functions to make each axis of BERT embeddings interpretable at both ends (Engler et al., 2022; Mathew et al., 2020).

However, our study takes a distinct approach. We do not rely on explicit constraints utilized in the aforementioned methods. Instead, we leverage transformations based on the inherent information within the embeddings. Our motivation aligns with that of Park et al. (2017) and Musil and Mareček (2022), aiming to incorporate interpretability into each axis of word vectors. Similar to previous studies (Honkela et al., 2010; Musil and Mareček, 2022), we have confirmed that interpretable axes are found by applying ICA to a set of embeddings.

### 2.2 Cross-lingual embeddings

**Cross-lingual mapping.** To address the task of cross-lingual alignment, numerous methodologies have been introduced to derive cross-lingual mappings. Supervised techniques that leverage translation pairs as training data have been proposed, such as the linear transformation approach (Mikolov et al., 2013b). Studies by Xing et al. (2015) and Artetxe et al. (2016) demonstrated enhanced performance when constraining the transformation matrices to be orthogonal. Furthermore, Artetxe et al. (2017) proposed a method for learning transformations from a minimal data set. As for unsupervised methods that do not leverage translation pairs for training, Lample et al. (2018) proposed an approach incorporating adversarial learning, while Artetxe et al. (2018) introduced a robust self-learning method. Additionally, unsupervised methods employing optimal transportation have been presented: Alvarez-Melis and Jaakkola (2018) introduced a method utilizing the Gromov-Wasserstein distance, while studies by Grave et al. (2019) and Aboagye et al. (2022) suggested methods that employ the Wasserstein distance.

**Multilingual language models.** Studies have demonstrated that a single BERT model, pretrained with a multilingual corpus, acquires cross-lingual grammatical knowledge (Pires et al., 2019; Chi et al., 2020). Further research has also been conducted to illustrate how such multilingual models express cross-lingual knowledge through embeddings (Chang et al., 2022).

**Our approach.** These cross-lingual studies, even the 'unsupervised' mapping, utilize embeddings

from multiple languages for training. Unlike these, we apply ICA to each language individually and identify the inherent semantic structure in each without referencing other languages. Thus ICA, as well as PCA, is an unsupervised transformation in a stronger sense. In our study, the embeddings from multiple languages and the translation pairs are used solely to verify that the identified semantic structure is shared across these languages. While it is understood from previous research that there exists a shared structure in embeddings among multiple languages (i.e., their shapes are similar), the discovery in our study goes beyond that by revealing the universal geometric patterns of embeddings with intrinsic interpretable axes (i.e., clarifying the shapes of embedding distributions; see Fig. 4).

## 3 ICA: Revealing the semantic structure in the geometric patterns of embeddings

We analyzed word embeddings using ICA. It was discovered that they possess inherent interpretability and sparsity. ICA can unveil these properties of embeddings.

### 3.1 PCA-transformed embeddings

Before explaining ICA, we briefly explain PCA, widely used for dimensionality reduction and whitening, or sphering, of feature vectors.

The pre-trained embedding matrix is represented as $\mathbf{X} \in \mathbb{R}^{n \times d}$, where $\mathbf{X}$ is assumed to be centered; it is preprocessed so that the mean of each column is zero. Here, $n$ represents the number of embeddings, and $d$ is the number of embedding dimensions. The $i$-th row of $\mathbf{X}$, denoted as $\mathbf{x}_i \in \mathbb{R}^d$, corresponds to the word vector of the $i$-th word, or the embedding computed by an image model from the $i$-th image.

In PCA, the transformed embedding matrix $\mathbf{Z} \in \mathbb{R}^{n \times d}$ is computed using algorithms such as Singular Value Decomposition (SVD) of $\mathbf{X}$. This process identifies the directions that explain the most variance in the data. The transformation can be expressed using a transformation matrix $\mathbf{A} \in \mathbb{R}^{d \times d}$ as follows:

$$\mathbf{Z} = \mathbf{X}\mathbf{A}.$$

The columns of $\mathbf{Z}$ are called principal components. The matrix $\mathbf{Z}$ is *whitened,* meaning that each column has a variance of 1 and all the columns are uncorrelated. In matrix notation, $\mathbf{Z}^\top \mathbf{Z}/n = \mathbf{I}_d$, where $\mathbf{I}_d \in \mathbb{R}^{d \times d}$ represents the identity matrix.

### 3.2 ICA-transformed embeddings

In Independent Component Analysis (ICA), the goal is to find a transformation matrix $\mathbf{B} \in \mathbb{R}^{d \times d}$ such that the columns of the resulting matrix $\mathbf{S} \in \mathbb{R}^{n \times d}$ are as independent as possible. This transformation is given by:

$$\mathbf{S} = \mathbf{X}\mathbf{B}.$$

The columns of $\mathbf{S}$ are called independent components. The independence of random variables is a stronger condition than uncorrelatedness, and when random variables are independent, it implies that they are uncorrelated with each other. While both PCA and ICA produce whitened embeddings, their scatterplots appear significantly different, as observed in Fig. 5; refer to Appendix B for more details. While PCA only takes into account the first and second moments of random variables (the mean vector and the variance-covariance matrix), ICA aims to achieve independence by incorporating the third moment (skewness), the fourth moment (kurtosis) and higher-order moments through non-linear contrast functions (Fig. 6).

In the implementation of FastICA[1], PCA is used as a preprocessing step for computing $\mathbf{Z}$, and

$$\mathbf{S} = \mathbf{Z}\mathbf{R}_{\text{ica}}$$

is actually computed[2], and we seek an orthogonal matrix $\mathbf{R}_{\text{ica}}$ that makes the columns of $\mathbf{S}$ as independent as possible (Hyvärinen and Oja, 2000). The linear transformation with an orthogonal matrix involves only rotation and reflection of the $\mathbf{z}_i$ vectors, ensuring that the resulting matrix $\mathbf{S}$ is also whitened, meaning that the embeddings of ICA, as well as those of PCA, are isotropic with respect to the variance-covariance matrix (Appendix A).

According to the central limit theorem, when multiple variables are added and mixed together, they tend to approach a normal distribution. Therefore, in ICA, an orthogonal matrix $\mathbf{R}_{\text{ica}}$ is computed to maximize a measure of non-Gaussianity for each column in $\mathbf{S}$, aiming to recover independent variables (Hyvärinen and Oja, 2000). This idea is rooted in the notion of 'projection pursuit' (Huber, 1985), a long-standing idea in the field. Since the normal distribution maximizes entropy among probability distributions with fixed mean and variance, measures of non-Gaussianity are interpreted as approximations of neg-entropy.

---

[1] We used FastICA in Scikit-learn (Pedregosa et al., 2011).
[2] Since we can express $\mathbf{S} = \mathbf{X}\mathbf{A}\mathbf{R}_{\text{ica}}$, and thus specifying $\mathbf{R}_{\text{ica}}$ is equivalent to specifying $\mathbf{B} = \mathbf{A}\mathbf{R}_{\text{ica}}$.

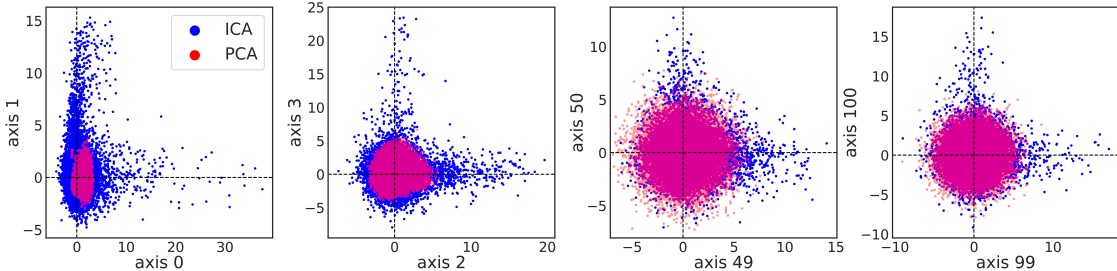

Figure 5: Scatterplots of word embeddings along specific axes: (0, 1), (2, 3), (49, 50), and (99, 100). The axes for ICA and PCA-transformed embeddings were arranged in descending order of skewness and variance, respectively.

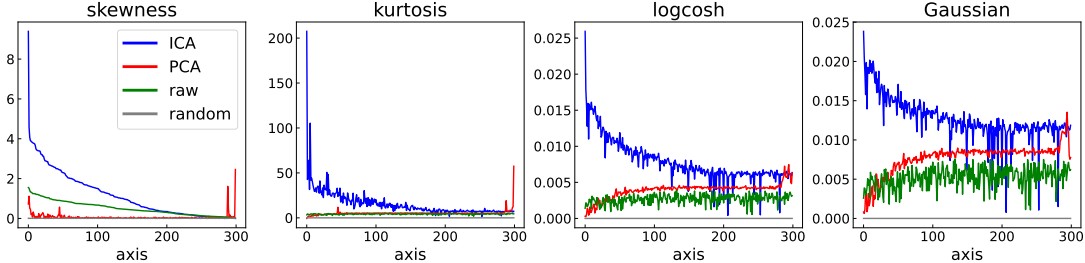

Figure 6: Measures of non-Gaussianity for each axis of ICA and PCA-transformed word embeddings. Additionally, the measures for each component of the raw word embedding and a Gaussian random variable are plotted as baselines. Larger values indicate deviation from the normal distribution. The axes found by ICA are far more non-Gaussian than those found by PCA. For more details, refer to Appendix B.

### 3.3 Interpretability and low-dimensionality of ICA-transformed embeddings

Fig. 1 illustrates that the key components of a word embedding are formed by specific axes that capture the meanings associated with each word. Specifically, this figure showcases five axes selected from the ICA-transformed word embeddings, that is, five columns of $\mathbf{S}$. The word embeddings $\mathbf{X}$ have a dimensionality of $d = 300$ and were trained on the text8 corpus using Skip-gram with negative sampling. For details of the experiment, refer to Appendix B.

**Interpretability.** Each of these axes represents a distinct meaning and can be interpreted by examining the top words based on their normalized component values. For example, words like *dishes, meat, noodles* have high values on axis 16, while words like *cars, car, ferrari* have high values on axis 26. We labeled each axis with the word having the highest component value, enclosed in brackets like *[dishes]* for axis 16, and *[cars]* for axis 26.

**Low-dimensionality.** The meaning of a word is approximately represented by the combination of a few axes. For example, the word *ferrari* has large values on *[cars]* (axis 26) and *[italian]* (axis 34). This indicates that the meaning of the word *ferrari*

is approximately represented by these two axes. Quantitative evaluation is provided in Section 6.

## 4 Universality across languages

This section examines the results of conducting ICA on word embeddings, each trained individually from different language corpora. Interestingly, the meanings of the axes discovered by ICA appear to be the same across all languages. For a detailed description of the experiment, refer to Appendix C.

**Setting.** We utilized the fastText embeddings by Grave et al. (2018), each trained individually on separate corpora for 157 languages. In this experiment, we used seven languages: English (EN), Spanish (ES), Russian (RU), Arabic (AR), Hindi (HI), Chinese (ZH), and Japanese (JA). The dimensionality of each embedding is $d = 300$. For each language, we selected $n = 50,000$ words, and computed the PCA-transformed embeddings $\mathbf{Z}_{\text{lang}}$ and the ICA-transformed embeddings $\mathbf{S}_{\text{lang}}$ for each of the seven centered embedding matrices $\mathbf{X}_{\text{lang}}$ ($\text{lang} \in \{\text{EN}, \text{ES}, \text{RU}, \text{AR}, \text{HI}, \text{ZH}, \text{JA}\}$).

We then performed the permutation of axes to find the best alignment of axes between languages. The matching is measured by the cross-correlation coefficients between languages.

**Results.** The upper panels of Fig. 3 display the cross-correlation coefficients between the first 100 axes of English and those of Spanish embeddings. The significant diagonal elements and negligible off-diagonal elements observed in Fig. 3a suggest a strong alignment of axes, indicating a good correspondence between the ICA-transformed embeddings. Conversely, the less pronounced diagonal elements and non-negligible off-diagonal elements observed in Fig. 3b indicate a less favorable alignment between the PCA-transformed embeddings.

The semantic axes identified by ICA, referred to as 'independent semantic axes' or 'independent semantic components', appear to be nearly universal, regardless of the language. In Fig. 2a, heatmaps visualize the ICA-transformed embeddings. The upper panels display the top 5 words for each of the first 100 axes in English and their corresponding translations in the other languages. It is evident that words sharing the same meaning in different languages are represented on the corresponding axes. The lower panels show the first 5 axes with their corresponding words. ICA identified axes in each language related to *first names, ships-and-sea, country names, plants*, and *meals* as independent semantic components. Overall, the heatmaps for all languages exhibit very similar patterns, indicating a shared set of independent semantic axes in the ICA-transformed embeddings across languages.

## 5 Universality in algorithm and modality

We expand the analysis from the previous section to two additional settings. The first setting involves comparing fastText and BERT, while the second setting involves comparing multiple image models and fastText simultaneously.

### 5.1 Contextualized word embeddings

**Setting.** Sentences included in the One Billion Word Benchmark (Chelba et al., 2014) were processed using a BERT-base model to generate contextualized embeddings for $n = 100,000$ tokens, each with a dimensionality of $d = 768$. We computed PCA and ICA-transformed embeddings for both the BERT embeddings and the English fastText embeddings. As with the cross-lingual case of Section 4, the axes are aligned between BERT and fastText embeddings by permuting the axes based on the cross-correlation coefficients. Further details can be found in Appendix D.1.

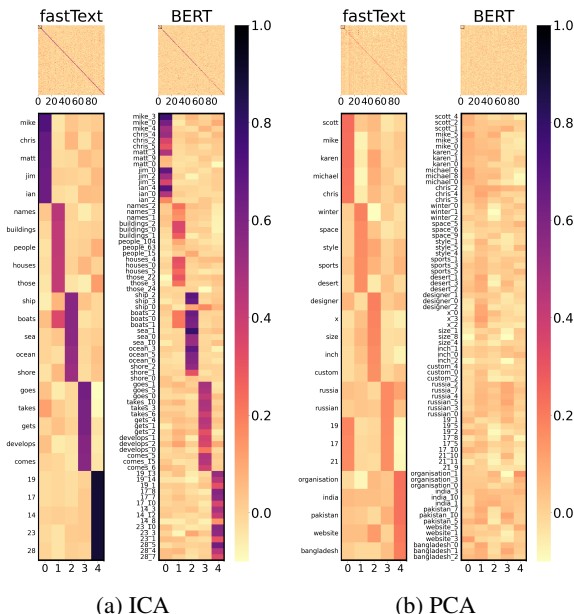

(a) ICA    (b) PCA

Figure 7: The rows represent normalized word embeddings for English fastText and English BERT transformed by (a) ICA and (b) PCA. The components from the first 100 axes of 500 fastText and 1,500 BERT embeddings are displayed in the upper panels, and the first five axes are magnified in the lower panels. For each axis of fastText embeddings, the top 5 words were chosen based on their component values. For each of these words, 3 corresponding tokens from BERT were chosen randomly. Correlation coefficients between transformed axes were utilized to establish axis correspondences and carry out axis permutations, as detailed in Section 5.1 and Appendix D.1.

**Results.** In Fig. 7a, the heatmaps for fastText and BERT exhibit strikingly similar patterns, indicating a shared set of independent semantic axes in the ICA-transformed embeddings for both fastText and BERT. The lower heatmaps show the first five axes with meanings *first names, community, ships-and-sea, verb*, and *number*. Furthermore, the middle panel in Fig. 3a demonstrates a good alignment of axes between fastText and BERT embeddings. On the other hand, PCA gives a less favorable alignment, as seen in Figs. 3b and Fig. 7b.

### 5.2 Image embeddings

**Setting.** We randomly sampled images from the ImageNet dataset (Russakovsky et al., 2015), which consists of 1000 classes. For each class, we collected 100 images, resulting in a total of $n = 100,000$ images. These images were passed through the five pre-trained image models listed in Table 1, and we obtained embeddings from the layer just before the final layer of each model.

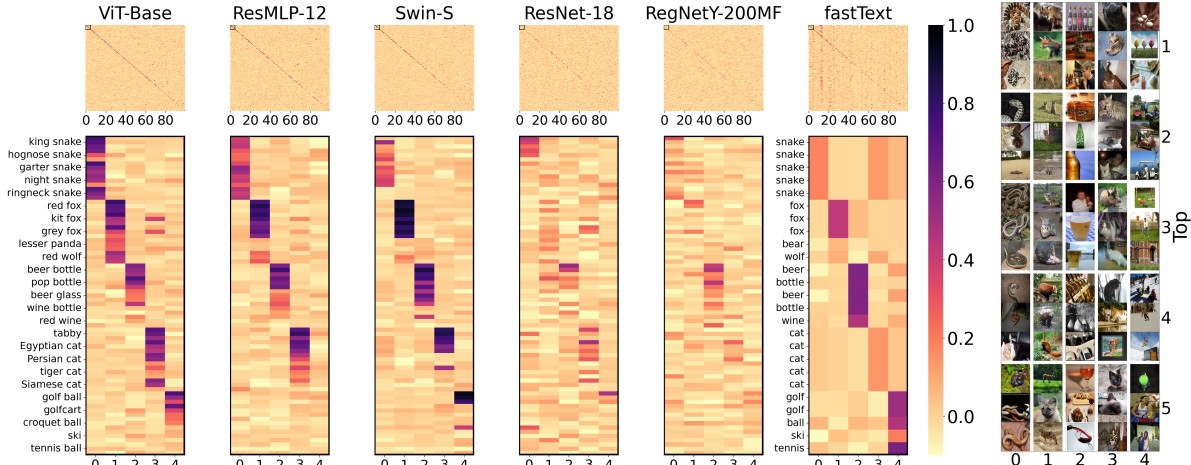

(a) Normalized ICA-transformed embeddings of images and words for five image models and English fastText.

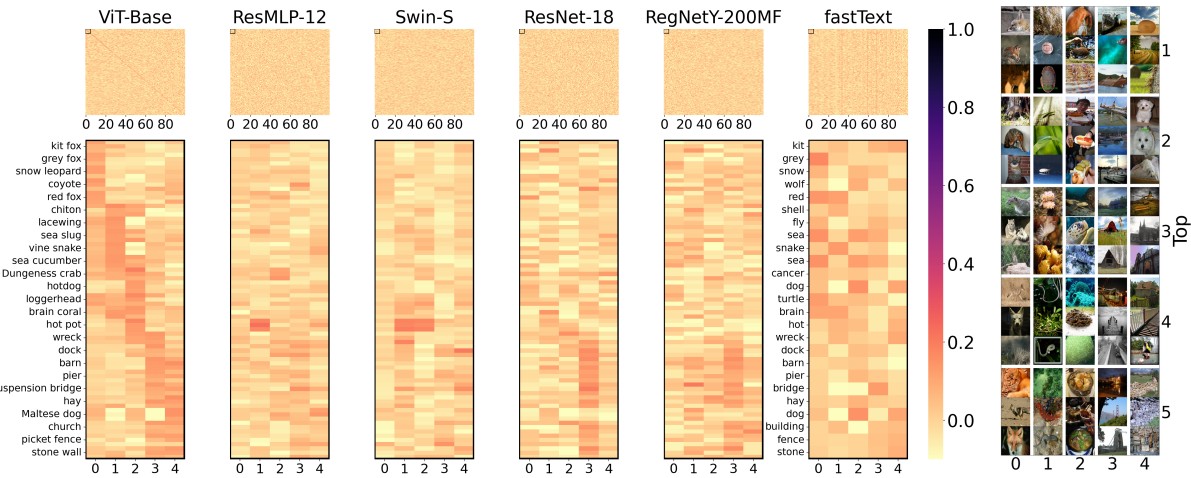

(b) Normalized PCA-transformed embeddings of images and words for five image models and English fastText.

Figure 8: The rows represent (a) ICA-transformed embeddings and (b) PCA-transformed embeddings of images or words. The lower panels magnify the first five axes. The components from the first 100 axes are displayed for the 1500 image embeddings and 500 word embeddings. For each axis of ViT-Base embeddings, the top 5 ImageNet classes were chosen based on the mean of their component values, and 3 images were sampled randomly from each class. These images were also used for the other image models. For fastText, words that best describe the ImageNet class name were selected. Correlation coefficients between transformed axes were utilized to establish axis correspondences and carry out axis permutations; ViT-Base was used as the reference model among the image models to match the axes, and then the axes were aligned between ViT-Base and English fastText to ensure the correlation coefficients are in descending order, as detailed in Section 5.2 and Appendix D.2.

| model | weight | $d$ |
|---|---|---|
| ViT-Base | vit_base_patch32_224_clip_laion2b | 768 |
| ResMLP-12 | resmlp_12_224 | 384 |
| Swin-S | swin_small_patch4_window7_224 | 768 |
| ResNet-18 | resnet18 | 512 |
| RegNetY-200MF | regnety_002 | 368 |

Table 1: pre-trained image models.

Among these models, we selected a specific model of ViT-Base (Dosovitskiy et al., 2021) as our reference image model. This particular ViT-Base model was trained with a focus on aligning with text em-

beddings (Radford et al., 2021). We computed PCA and ICA-transformed embeddings for the five image models. As with the cross-lingual case of Section 4, the axes are aligned between ViT-Base and each of the other four image models by permuting the axes based on the cross-correlation coefficients. Additionally, to align the axes between ViT-Base and English fastText, we permuted the axes based on the cross-correlation coefficients that were computed using ImageNet class names and fastText vocabulary as a means of linking the two modalities. Further details can be found in Appendix D.2.

**Results.** In Fig. 8a, the heatmaps of the five image models and fastText exhibit similar patterns, indicating a shared set of independent semantic axes in the ICA-transformed embeddings for both images and words. While we expected a good alignment between ViT-Base and fastText, it is noteworthy that we also observe a good alignment of ResMLP (Touvron et al., 2023) and Swin Transformer (Liu et al., 2021) with fastText. Furthermore, Fig. 3a demonstrates a very good alignment of axes between ViT-Base and ResMLP. This suggests that the independent semantic components captured by ICA are not specific to a particular image model but are shared across multiple models. On the other hand, PCA gives a less favorable alignment, as seen in Figs. 3b and 8b.

## 6 Quantitative evaluation

We quantitatively evaluated the interpretability (Section 6.1) and low-dimensionality (Section 6.2) of ICA-transformed embeddings comparing with other whitening methods (PCA, ZCA) as well as a well-known rotation method (varimax). These baseline methods are described in Appendix E.1. In the monolingual experiments, we used 300-dimensional word embeddings trained using the SGNS model on the text8 corpus, as outlined in Appendix B. Furthermore, we assessed the cross-lingual performance (Section 6.3) of PCA and ICA-transformed embeddings, along with two other supervised baseline methods.

### 6.1 Interpretability: word intrusion task

We conducted the word intrusion task (Sun et al., 2016; Park et al., 2017) in order to quantitatively evaluate the interpretability of axes. In this task, we first choose the top $k$ words from each axis, and then evaluate the consistency of their word meaning based on the identifiability of the intruder word. For instance, consider a word group of $k = 5$, namely, {*windows, microsoft, linux, unix, os*} with the consistent theme of operating systems. Then, *hamster* should be easily identified as an intruder. Details are presented in Appendix E.3.

**Results.** The experimental results presented in Table 2 show that the top words along the axes of ICA-transformed embeddings exhibit more consistent meanings compared to those of other methods. This confirms the superior interpretability of axes in ICA-transformed embeddings.

|  | ZCA | PCA | Varimax | ICA |
|---|---|---|---|---|
| DistRatio | 1.04 | 1.13 | 1.26 | **1.57** |

Table 2: A large value of DistRatio indicates the consistency of word meaning in the word intrusion task. We set $k = 5$, and the reported score is the average of 10 runs with randomly selected intruder words.

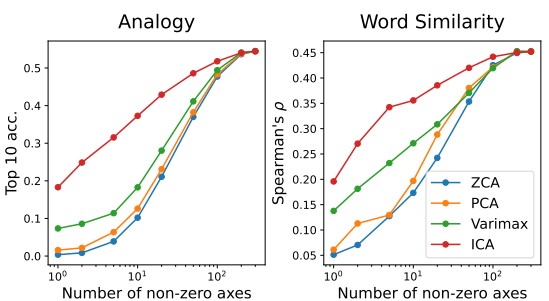

Figure 9: The performance of several whitened word embeddings when reducing the non-zero components. The values are averaged over 14 analogy tasks or six word similarity tasks. The performance of a specific case in analogy and word similarity tasks is shown in Fig. 17 in Appendix E.4.

### 6.2 Low-dimensionality: analogy task & word similarity task

We conducted analogy tasks and word similarity tasks using a reduced number of components from the transformed embeddings. Specifically, we evaluated how well the transformed embedding retains semantic information even when reducing the non-zero components from the least significant ones. For each whitened embedding, we retained the $k$ most significant components unchanged while setting the remaining components to zero. The specific axes we used depend on each embedding. The performance was evaluated for the number of axes $k$ ranging from 1 to 300.

**Results.** Fig. 9 demonstrates that the ICA-transformed embedding has the highest average performance throughout the entire dataset. Detailed settings and results, including those for unwhitening cases, are presented in Appendix E.4. These experimental results show that the ICA-transformed embedding effectively represents word meanings using only a few axes.

### 6.3 Universality: cross-lingual alignment

In Fig. 2a, we visually inspected the cross-lingual alignment obtained by permutating ICA-transformed embeddings, and we observed remarkably good alignment across languages. In this sec-

| Dataset | Method | Original | Rand. |
|---|---|---|---|
| 157langs-fastText | LS | 84.72 | 55.28 |
| | Proc | 84.95 | 18.72 |
| | ICA | 19.48 | 18.60 |
| | PCA | 0.92 | 0.64 |
| MUSE-fastText | LS | 78.91 | 51.44 |
| | Proc | 78.41 | 20.19 |
| | ICA | 43.27 | 43.31 |
| | PCA | 2.13 | 0.40 |

Table 3: The average top-1 accuracy of the cross-lingual alignment task from English to other languages. Two datasets of fastText embeddings (157langs and MUSE) were evaluated with the two types of embeddings (Original and Random-transformation). LS and Proc are supervised transformations using both the source and target embeddings, while ICA and PCA are unsupervised transformations. For the complete results, refer to Table 15 in Appendix E.5.

tion, we go beyond that by thoroughly examining the performance of the alignment task. Details are presented in Appendix E.5.

**Datasets.** In addition to '157langs-fastText' used in Section 4 (Grave et al., 2018), we used 'MUSE-fastText', pre-aligned embeddings across languages (Lample et al., 2018). The source language is English (EN), and the target languages are Spanish (ES), French (FR), German (DE), Italian (IT), and Russian (RU). For each language, we selected 50,000 words.

**Supervised baselines.** We used two supervised baselines that learn a transformation matrix $\mathbf{W} \in \mathbb{R}^{d \times d}$ by leveraging both the source embedding $\mathbf{X}$ and the target embedding $\mathbf{Y}$. Each row of $\mathbf{X}$ and $\mathbf{Y}$ corresponds to a translation pair. We minimized $\|\mathbf{X}\mathbf{W} - \mathbf{Y}\|_2^2$ by the least squares (LS) method (Mikolov et al., 2013b) or the Procrustes (Proc) method with the constraint that $\mathbf{W}$ is an orthogonal matrix (Xing et al., 2015; Artetxe et al., 2016).

**Random transformation.** In addition to the original fastText embeddings, we considered embeddings transformed by a random matrix $\mathbf{Q} \in \mathbb{R}^{d \times d}$ that involves random rotation and random scaling. Specifically, for each $\mathbf{X}$, we independently generated $\mathbf{Q}$ to compute $\mathbf{X}\mathbf{Q}$.

**Results.** Table 3 shows the top-1 accuracy of the cross-lingual alignment task. The values are averaged over the five target languages.

LS consistently performed the best, or nearly the best, across all the settings because it finds the optimal mapping from the source language to the target language by leveraging both the embeddings as well as translation pairs. Therefore, we consider LS as the reference method in this experiment. Proc performed similarly to LS in the original embeddings, but its performance deteriorated with the random transformation. The original word embeddings had very similar geometric arrangements across languages, but the random transformation distorted the arrangements so that the orthogonal matrix in Proc was not able to recover the original arrangements.

ICA generally performed well, despite being an unsupervised method. In particular, ICA was not affected by random transformation and performed as well as or better than Proc. The higher performance of ICA for MUSE than for 157langs is likely due to the fact that MUSE is pre-aligned. On the other hand, PCA performed extremely poorly in all the settings. This demonstrates the challenge of cross-lingual alignment for unsupervised transformations and highlights the superiority of ICA.

The observations from this experiment can be summarized as follows. ICA was able to identify independent semantic axes across languages. Furthermore, ICA demonstrated robust performance even when the geometric arrangements of embeddings were distorted. Despite being an unsupervised transformation method, ICA achieved impressive results and performed comparably to the supervised baselines.

# 7 Conclusion

We have clarified the universal semantic structure in the geometric patterns of embeddings using ICA by leveraging anisotropic distributions remaining in the whitened embeddings. We have verified that the axes defined by ICA are interpretable and that embeddings can be effectively represented in low-dimensionality using a few of these components. Furthermore, we have discovered that the meanings of these axes are nearly universal across different languages, algorithms, and modalities. Our findings are supported not only by visual inspection of the embeddings but also by quantitative evaluation, which confirms the interpretability, low-dimensionality, and universality of the semantic structure. The results of this study provide new insights for pursuing the interpretability of models. Specifically, it can lead to the creation of interpretable models and the compression of models.

## Limitations

- Due to the nature of methods that identify semantic axes by linearly transforming embeddings, the number of independent semantic components is limited by the dimensionality of the original embeddings.

- ICA transforms the data in such a way that the distribution of each axis deviates from the normal distribution. Therefore, ICA is not applicable if the original embeddings follow a multivariate normal distribution. However, no such issues were observed for the embeddings considered in this paper.

- In Section 4, we utilized translation pairs to confirm the shared semantic structure across languages. Consequently, without access to such translation pairs, it becomes infeasible to calculate correlation coefficients and achieve successful axis matching. Therefore, in the future, we intend to investigate whether it is possible to perform matching by comparing distributions between axes using optimal transport or other methods without relying on translation pairs.

- When comparing heatmaps of embeddings across languages in Fig. 2, we looked at five words from each axis. Thus only a small fraction of vocabulary words were actually examined for verifying the shared structure of geometric patterns of embeddings. Although this issue is already compensated by the plot of cross-correlations in Fig. 3, where a substantial fraction of vocabulary words were examined, we seek a better way to verify the shared structure in future work. For example, the scatter plots in Fig. 15 may help us understand the entire structure of word embedding distributions.

## Ethics Statement

This study complies with the ACL Ethics Policy.

## Acknowledgements

We would like to thank Sho Yokoi for the discussion and anonymous reviewers for their helpful advice. This study was partially supported by JSPS KAKENHI 22H05106, 23H03355, JST CREST JPMJCR21N3.

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

## A Whitening and isotropic embeddings

In addition to the whitened embeddings $\mathbf{Z}$ obtained through PCA, we also consider various other whitened embeddings (Kessy et al., 2018). They can be represented in the form of linear transformations using an orthogonal matrix $\mathbf{R}$:

$$\mathbf{Y} = \mathbf{ZR}.$$

These transformed embeddings $\mathbf{Y}$ with any $\mathbf{R}$ are again whitened[3], meaning that the embeddings are isotropic with respect to the variance-covariance matrix. Also, the centering step of whitening makes the embeddings centered, and the transformed embeddings $\mathbf{Y}$ with any $\mathbf{R}$ are again centered[4], meaning that the embeddings are isotropic with respect to the mean vector. However, the linear transformation cannot make the embeddings isotropic with respect to the third and higher-order moments.

The row vectors $\mathbf{y}_i = (y_{i1}, \ldots, y_{id})$ and $\mathbf{z}_i = (z_{i1}, \ldots, z_{id})$ of $\mathbf{Y}$ and $\mathbf{Z}$, respectively, satisfy the equation[5]:

$$\langle \mathbf{y}_i, \mathbf{y}_j \rangle = \langle \mathbf{z}_i, \mathbf{z}_j \rangle,$$

where $\langle \mathbf{a}, \mathbf{b} \rangle = \sum_{k=1}^{d} a_k b_k$ represents the inner product. Therefore, the inner products of embeddings are preserved under this transformation, indicating that the performance of tasks based on inner products, such as those using cosine similarity, remains unchanged.

## B Details of experiment in Section 3

We summarize the details of the embeddings used in Figure 1 and the monolingual quantitative evaluations in Sections 6.1 and 6.2.

**Corpus.** We used the text8 (Mahoney, 2011), which is an English corpus data with the size of $N = 17.0 \times 10^6$ tokens and $|V| = 254 \times 10^3$ vocabulary words. We used all the tokens separated by spaces. The frequency of word $w \in V$ in the corpus is denoted as $p(w)$, where $\sum_{w \in V} p(w) = 1$.

**Training of the SGNS model.** Word embeddings were trained[6] by optimizing the same objective

---

[3]In general, for an orthogonal matrix $\mathbf{R}$, i.e., $\mathbf{R}^\top \mathbf{R} = \mathbf{I}_d$, consider a transformed matrix $\mathbf{Y} = \mathbf{ZR}$. Then $\mathbf{Y}$ is whitened, because $\mathbf{Y}^\top \mathbf{Y}/n = (\mathbf{ZR})^\top \mathbf{ZR}/n = \mathbf{R}^\top \mathbf{Z}^\top \mathbf{ZR}/n = \mathbf{R}^\top \mathbf{R} = \mathbf{I}_d$.

[4]For centered embeddings $\mathbf{Z}$, the mean vector is $(\mathbf{1}_n/n)^\top \mathbf{Z} = \mathbf{0}_d^\top$, where $\mathbf{1}_n = (1, \ldots, 1)^\top \in \mathbb{R}^n$ and $\mathbf{0}_d = (0, \ldots, 0)^\top \in \mathbb{R}^d$. Then, for any orthogonal matrix $\mathbf{R}$, $(\mathbf{1}_n/n)^\top \mathbf{Y} = (\mathbf{1}_n/n)^\top \mathbf{ZR} = \mathbf{0}_d^\top \mathbf{R} = \mathbf{0}_d^\top$.

[5]Since $\mathbf{y}_i = \mathbf{z}_i \mathbf{R}$, we have $\langle \mathbf{y}_i, \mathbf{y}_j \rangle = \mathbf{y}_i \mathbf{y}_j^\top = \mathbf{z}_i \mathbf{R}(\mathbf{z}_j \mathbf{R})^\top = \mathbf{z}_i \mathbf{R}\mathbf{R}^\top \mathbf{z}_j^\top = \mathbf{z}_i \mathbf{z}_j^\top = \langle \mathbf{z}_i, \mathbf{z}_j \rangle$.

[6]We used AMD EPYC 7702 64-Core Processor (64 cores × 2). In this setting, the CPU time is estimated at about 12 hours.

function used in Mikolov et al. (2013c). Parameters used to train SGNS are summarized in Table 4. The learning rate shown is the initial value, which we decreased linearly to the minimum value of $1.0 \times 10^{-4}$ during the learning process. The negative sampling distribution was proportional to the 3/4-th power of the word frequency, $p(w)^{3/4}$. The elements of $\mathbf{x}_i$ were initialized by the uniform distribution over $[-0.5, 0.5]$ divided by the dimensionality of the embedding, and the elements of $\mathbf{x}'_i$ were initialized by zero.

| | |
|---|---|
| Dimensionality | 300 |
| Epochs | 100 |
| Window size $h$ | 10 |
| Negative samples $\nu$ | 5 |
| Learning rate | 0.025 |
| Min count | 1 |

Table 4: SGNS parameters.

**Discarding low-frequency words.** After computing the embeddings, we discarded words $w$ that appeared less than 10 times in the text8 corpus. The new vocabulary $V'$ consists of the 47,134 words that appeared 10 times or more in the text8 corpus.

**Resampling word embeddings.** We resampled the word $w$ with probability $q(w)$ when preparing the data matrix $\mathbf{X} \in \mathbb{R}^{n \times d}$. As an example, we will explain the procedure when employing $q(w) \propto p(w)$ for $w \in V'$. First, we randomly sampled 100,000 words from $V'$ with replacement using the weight $q(w)$. This resulted in the selection of 14,942 unique words. Subsequently, we added 15,058 words from the remaining unselected words, ordered by descending frequency, to reach a total of 30,000 unique words. Each row of $\mathbf{X}$ represents the embeddings of the $n = 115{,}058$ words selected through the aforementioned process.

**Selection of resampling weight.** We considered resampling weights in the form of $q(w) \propto p(w)^{\alpha}$, where $\alpha$ takes values from the candidate set $\alpha \in \{1/2, 3/4, 1\}$. We conducted an experiment to determine the optimal value of $\alpha$. For each $q(w)$, we prepared the data matrix $\mathbf{X}$ using the resampling method explained above. Additionally, we created an unweighted $\mathbf{X}$ with $n = |V'|$. We then computed ICA-transformed embeddings and evaluated the performance on the analogy and word similarity tasks using a reduced number of components, which are explained in detail in Section 6.2 and Appendix E.4. The results are shown in Fig. 10. We

observed that either $\alpha = 3/4$ or $\alpha = 1$ is the best, and we decided that using $\alpha = 1$ is appropriate for ease of implementation in general.

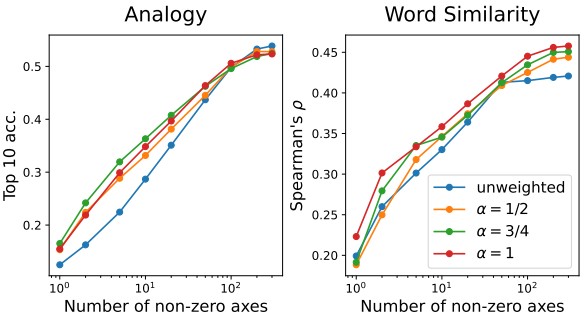

Figure 10: Performance of ICA-transformed embeddings using a reduced number of components with varying resampling weights specified by $\alpha$. Larger values indicate better performance. We measured the top-10 accuracy for the analogy task and the Spearman rank correlation for the word similarity task. These values represent averages across 14 analogy tasks and six word similarity tasks.

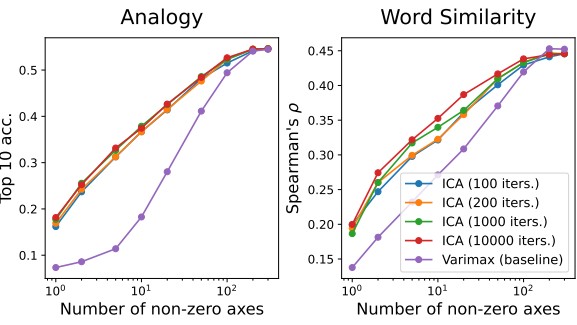

Figure 11: Performance of ICA-transformed embeddings with varying iterations. The settings are the same as those in Fig. 10.

**Selection of the number of FastICA iterations.** We demonstrate that our analysis remains unaffected by changes in the number of iterations in FastICA. Specifically, we evaluated the embeddings obtained by varying the number of iterations (100, 200, 1000, 10000) in FastICA. The evaluation was performed on the word intrusion task, analogy task, and word similarity task. The results are presented in Table 5 and Fig. 11. In both experiments, we observed that reducing the number of FastICA iterations slightly diminished task performance, although the difference was very small. Therefore, we conclude that changing the number of FastICA iterations did not significantly impact the results.

**ICA-transformed embeddings.** We utilized the implementation of FastICA, with the default setting

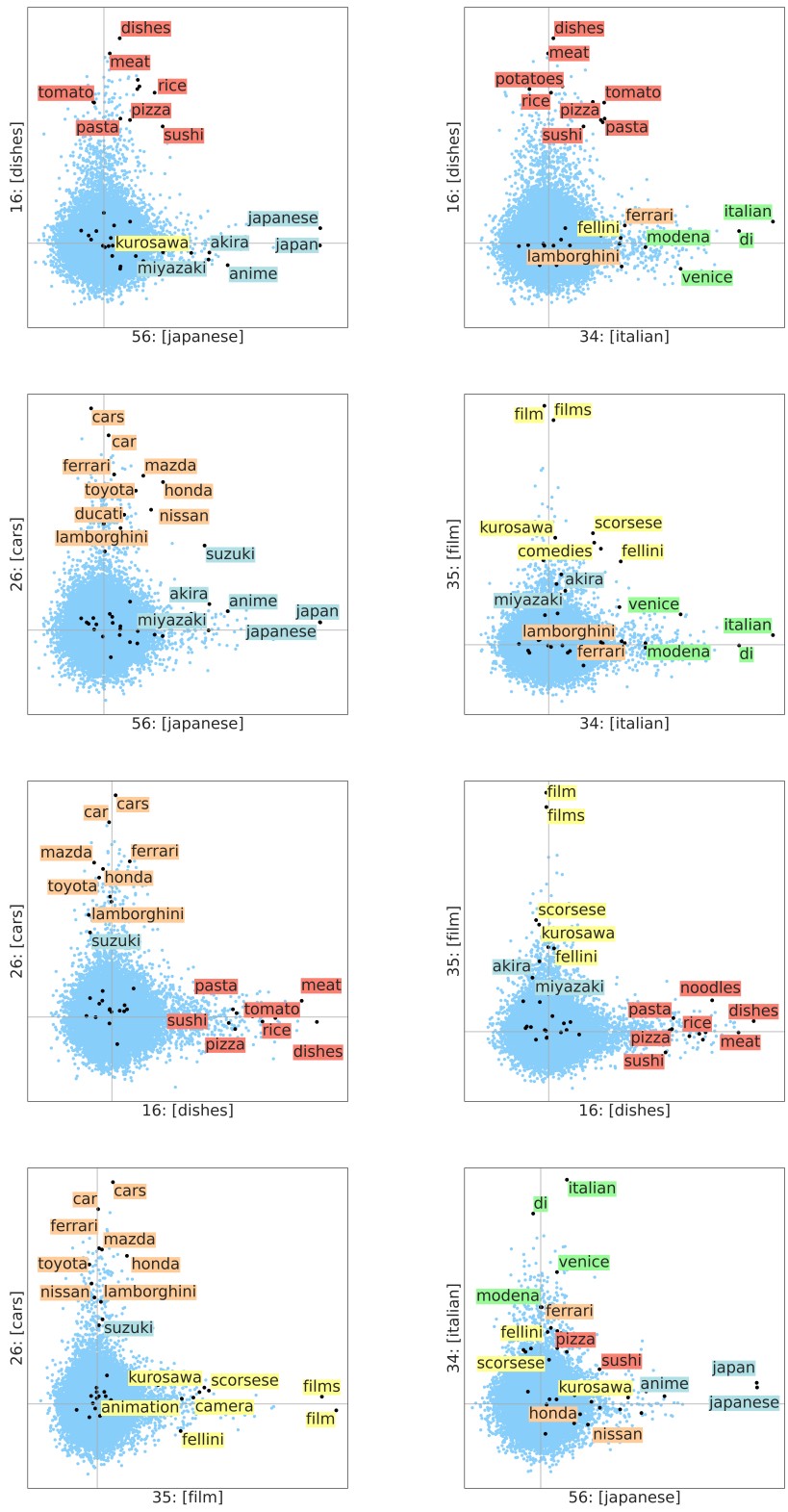

Figure 12: Scatter plots of the normalized ICA-transformed embedding for eight combinations of the five axes. The plots for the other two combinations are shown in Fig. 1. The words are highlighted with colors corresponding to their respective axes. We observe that each axis can be interpreted by words that have large values along that axis. Some words are represented by a combination of axes, such as *sushi = [japanese] + [dishes]* or *fellini = [italian] + [film]*. In some pairs of two axes, there may be no words represented by their combination. For example, in the plot of (*[dishes]*, *[cars]*) axes, no words are found where both of these components are large.

| | Varimax (baseline) | ICA (100 iterations) | ICA (200 iterations) | ICA (1000 iterations) | ICA (10000 iterations) |
|---|---|---|---|---|---|
| DistRatio | 1.26 | 1.42 | 1.47 | 1.52 | 1.55 |

Table 5: Performance of ICA-transformed embeddings with varying iterations. Large values of DistRatio indicate better interpretability. The settings are the same as those in Table 2.

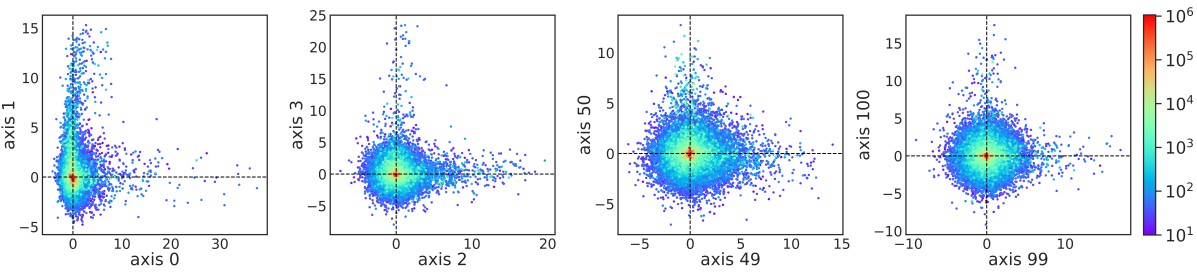

(a) ICA-transformed word embeddings. The axis numbers are sorted by skewness.

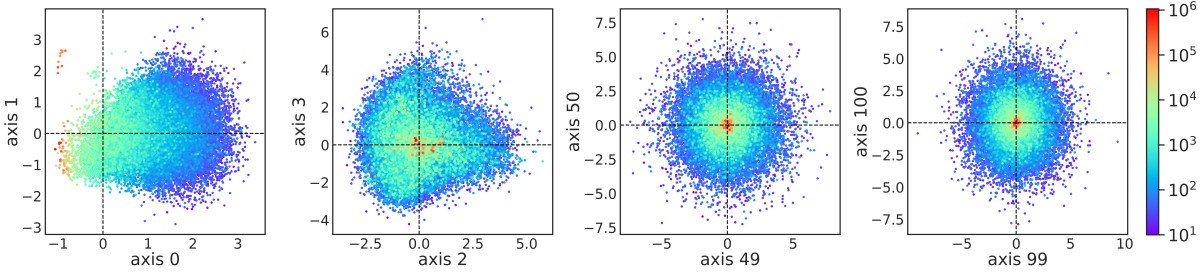

(b) PCA-transformed word embeddings. The axis numbers are sorted by variance.

Figure 13: Scatter plots of transformed word embeddings along specific axes: (0, 1), (2, 3), (49, 50), and (99, 100). Colors indicate word frequency in the corpus, with warmer colors being more frequent.

except for the number of iterations set to 10,000 and a tolerance of $10^{-10}$. The contrast function used was $G(x) = \log \cosh(x)$. It should be noted that the embeddings obtained through ICA have arbitrariness in the sign of each axis and the order of the axes. We calculated the skewness of each axis and flipped the sign of axes as necessary to ensure a positive sign of skewness. We then sorted the axes in descending order of skewness. When visualizing embeddings or selecting word sets, we normalized each embedding to have a unit norm for facilitating interpretation unless otherwise specified.

**Details of Fig. 1 in Section 1.** To illustrate the additive compositionality of embeddings, the words in the heatmap were selected as follows. For each of the six combinations of axes {*[dishes], [cars], [films]*} × {*[italian], [japanese]*}, we selected 20 words with the largest sum of the two component values. From these 20 words, we chose the top five words based on the second-largest component value among the five axes. As a result, a total of $6 \times 5 = 30$ words were selected in this process.

Next, we created scatterplots of the normalized ICA-transformed embeddings for all the ten combinations of two axes selected from {*[dishes], [cars], [films], [italian], [japanese]*}. Two of these scatterplots are shown in Fig. 1, and the remaining eight are presented in Fig. 12.

**Scatter plots of ICA and PCA-transformed word embeddings.** To visualize the transformed embedding, scatterplots are displayed with selected pairs of axes. Fig. 13a shows the ICA-transformed embeddings given in Section 3.2, plotting specified columns of **S**, with the axis numbers sorted in descending order of skewness. Fig. 13b shows the PCA-transformed embeddings given in Section 3.1, plotting specified columns of **Z**, with the axis numbers sorted in descending order of variance. Colors indicate word frequency in the corpus, with warmer colors being more frequent. Unlike other visualizations, each embedding is *not* normalized to have a unit norm. The plots in Fig. 5 in Sectoin 3 are superpositions of the plots in Figs. 13a and 13b, but the frequency information was omitted.

The distributions of these two types of word embedding have exactly the same shape in $\mathbb{R}^d$ because they can be transformed into each other by rotation as $\mathbf{S} = \mathbf{Z}\mathbf{R}_{\text{ica}}$. However, there are significant differences in the word distributions on each axis. The distribution of ICA-transformed embeddings exhibits anisotropy with a heavy-tailed shape along each axis, thereby characterizing the meaning of each axis. The distribution of PCA-transformed embeddings is more isotropic and lacks specific characterization of each axis, except for the top few axes that encode word frequency as pointed out by Mu and Viswanath (2018). Interestingly, pronounced frequency bias is not observed in the axes of ICA-transformed embeddings.

The spiky shape of the distribution of word embeddings observed in Fig. 13a is illustrated in Fig. 4. As Figs. 2a and 3a suggest, the distribution of word embeddings across multiple languages has a nearly common shape, so they can be mapped by orthogonal transformations.

**Measures of non-Gaussianity.** Fig. 6 in Sectoin 3 displays four non-Gaussianity measures. Let $X$ denote a random variable representing the component on a specified axis, and let $Z$ denote a Gaussian random variable; here the symbols $X$ and $Z$ are not related to $\mathbf{X}$ and $\mathbf{Z}$. Since both PCA and ICA-transformed embeddings are whitened, we assume that $X$ and $Z$ have a mean of zero and a variance of one; $\mathbb{E}(X) = \mathbb{E}(Z) = 0$ and $\mathbb{E}(X^2) = \mathbb{E}(Z^2) = 1$, where $\mathbb{E}()$ denotes the expectation.

- The first measure is the skewness $\mathbb{E}(X^3)$. If it is negative, we flip the sign of the axis so that $\mathbb{E}(X^3) \geq 0$. Since $\mathbb{E}(Z^3) = 0$, a large value of skewness indicates a deviation from Gaussianity.

- The second measure is the kurtosis $\mathbb{E}(X^4) - \mathbb{E}(Z^4)$, where $\mathbb{E}(Z^4) = 3$. We observed that it is nonnegative for most of the components of embeddings, indicating that their distributions are more spread out with heavy tails than the normal distribution.

- The third measure labeled logcosh is defined as $\{\mathbb{E}(G(X)) - \mathbb{E}(G(Z))\}^2$ with the contrast function $G(x) = \log \cosh(x)$ and $\mathbb{E}(G(Z)) = 0.374567207491438$. This measure is used as the objective function in the implementation of FastICA.

- The fourth measure labeled Gaussian is $\{\mathbb{E}(G(X)) - \mathbb{E}(G(Z))\}^2$ with the contrast function $G(x) = -\exp(-x^2/2)$ and $\mathbb{E}(G(Z)) = -1/\sqrt{2}$.

Properties of the last two measures are well studied in Hyvarinen (1999).

## C   Details of experiment in Section 4

The procedure described for this cross-lingual experiment serves as a template for the other experiments in the following sections.

**Dataset.** We employed the fastText embeddings trained for 157 languages by Grave et al. (2018), referred to as '157langs-fastText' in this study. We also made use of the dictionaries provided by MUSE (Lample et al., 2018) to aid in the mapping of English words to their counterparts in other languages[7]. Given that the vocabulary of 157langs-fastText is substantial, containing 2,000,000 words, our initial step was to lowercase and select non-duplicate words included in both the English-to-other language and other language-to-English dictionaries provided by MUSE. These dictionaries contain 6,500 unique source words from the combined train and test sets. Subsequently, from the 2,000,000 words, those not yet chosen were selected based on their frequency. We determined the frequency using the Python package word-freq (Speer, 2022), which provides word frequencies across various languages. The vocabulary was then capped at 50,000 words for each language.

**PCA and ICA-transformed embeddings.** We then implemented PCA and ICA transformations on the fastText embeddings, each containing 50,000 words per language. We used PCA and FastICA in Scikit-learn (Pedregosa et al., 2011) with ICA performed for a maximum of 10,000 iterations. Finally, we computed the skewness for each axis and flipped the sign of the axis if necessary to ensure positive skewness. These PCA and ICA transformations were applied to each language individually to identify the inherent semantic structure within each language without referencing other languages. The following steps are devoted to verifying that the identified semantic structure in

---

[7]Here, we solely utilized the dictionaries provided by MUSE and did not make use of the embeddings included in MUSE. The multi-lingual embeddings in MUSE are pre-aligned across languages, which makes them unsuitable for this particular experiment.

the axes of ICA-transformed embeddings is shared across these languages.

| Dataset | | ES | RU | AR | HI | ZH | JA |
|---|---|---|---|---|---|---|---|
| | Pairs | 11965 | 10883 | 11559 | 8689 | 8645 | 6992 |
| 157langs Train | Source | 4991 | 4996 | 4988 | 4983 | 4965 | 4821 |
| | Target | 10154 | 9512 | 9650 | 7149 | 6521 | 5715 |

Table 6: The number of translation pairs, unique source English words, and unique target words for each target language.

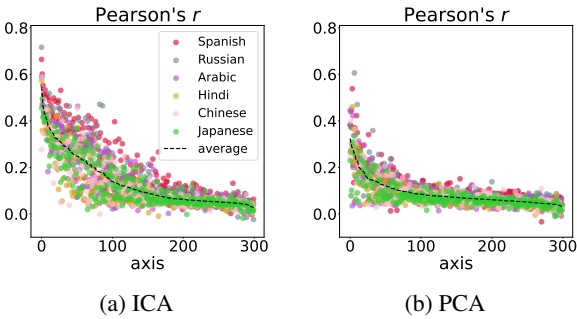

(a) ICA        (b) PCA

Figure 14: Correlation coefficients between the matched components of English and each of the other languages for ICA and PCA-transformed embeddings. The axes were sorted by the correlation coefficients averaged across the six languages.

**Word translation pairs.** As the first step of the verification, we established the correspondence between the embeddings of English and the embeddings of other languages in the 157langs-fastText dataset. This linking information between embeddings is necessary to compute cross-correlation coefficients. For the purpose of illustration, we will explain the procedure using the language pair of English and Spanish. First, we gathered all pairs of English and Spanish words from the train set of the MUSE dictionaries, including both the English-to-Spanish and Spanish-to-English translation pairs. Next, we filtered out any pairs where either the English word or the Spanish word was not included in the vocabulary set of 50,000 words prepared for each language. The total number of pairs collected was 11,965, where a word may be included in multiple translation pairs. The number of unique English words obtained from this process was 4,991. The results of applying this procedure to the six target languages are presented in Table 6, where the source language is English.

**Alignment of axes via permutation.** Next, we established a correspondence between the axes of

English word embeddings and those of other languages by appropriately permuting the indices of axes. This involves permuting the columns of the transformed embedding matrix. For illustrative purposes, we will explain the procedure using English and Spanish word embeddings. Since both the English and Spanish word embeddings have dimensions of 300, we computed a total of $300 \times 300$ cross-correlation coefficients using the translation pairs of English words and their corresponding Spanish words. From these cross-correlation coefficients, we identified matched pairs of axes with high correlations in a greedy fashion, starting from the highest correlation. The columns of Spanish word embeddings were then permuted to match those of English word embeddings. This procedure was applied to all other languages, ensuring that their axes align with those of English.

**Reordering axes.** Subsequently, we computed the average correlation coefficients of the aligned axes to determine the reordering of the axes based on their degree of similarity. For each language, we calculated 300 correlation coefficients between the axes and the corresponding axes of English. These correlation coefficients were then averaged across Spanish, Russian, Arabic, Hindi, Chinese, and Japanese. Finally, we reordered the axes in descending order based on the average correlation coefficient. The cross-correlations between the aligned axes as well as the average correlation coefficients are shown in Fig. 14.

**Diagnosing the axis alignment.** For both ICA-transformed and PCA-transformed embeddings, we demonstrated the $100 \times 100$ cross-correlation coefficients between the first 100 axes of reordered English and Spanish word embeddings in Fig. 3. The diagonal elements represent the correlation coefficients between the aligned pairs of axes, while the off-diagonal elements represent the correlation coefficients between unaligned axes. In the case of ICA-transformed embeddings, as depicted in Fig. 3a, it is clear that the diagonal elements exhibit significantly positive values, while the majority of the off-diagonal elements are close to zero. Thus, ICA gives a strong alignment of the axes. In contrast, for PCA-transformed embeddings, as shown in Fig. 3b, the diagonal elements are smaller compared to ICA, and a considerable number of off-diagonal elements deviate significantly from zero. Thus, PCA gives a less favorable alignment.

**Word selection for visualization.** After reordering the axes, we normalized all the embeddings to ensure that their norm is equal to 1. We focused on the first 100 axes for further analysis. We limited the selection to words that have translation pairs in the complete set of MUSE in all languages, including English, Spanish, Russian, Arabic, Hindi, Chinese, and Japanese. From this restricted set, we selected the top 5 words for each axis of the English word embeddings based on their largest component values. For the remaining languages, we selected words from the translation pairs. To ensure diversity, we excluded duplicates that occur in both singular and plural forms of nouns.

**Analyzing the heatmaps.** In Fig. 2, we presented heatmaps that illustrate the components of the normalized 500-word embeddings selected through this procedure for both ICA-transformed and PCA-transformed embeddings.

In Fig. 2a, which represents the ICA-transformed embeddings, we observe that the top 5 words for each axis have significant values along that axis. This leads to a diagonal pattern of large values due to the sparsity in other components. This consistent pattern is observed in all seven languages. The lower panels of the figure magnify the first five axes, allowing us to identify the top 5 words chosen for each axis. It is evident that each axis has a specific semantic relevance. For example, the first axis corresponds to *first names,* and we observe that such semantically meaningful axes can be effectively aligned across languages.

Conversely, in Fig. 2b representing the PCA-transformed embeddings, it can be observed that the top 500 words do not exhibit significant values along the diagonal. Additionally, in the magnified heatmaps depicting the 25 words, when compared to Fig. 2a, the semantics of the words for each axis appear to be more ambiguous.

**Visualization via scatterplots along the five axes.** In the heatmaps, we visualized embeddings for 25 words in each language. To overcome the limitation of the heatmap, which only allows us to view a small subset of words, we utilized scatterplots to visualize all the words in the vocabulary. In Fig. 15, we projected the normalized ICA-transformed embeddings into two dimensions using the same five axes as those used in the heatmaps.

Similarly to the heatmaps, the meaning of each axis can be interpreted based on the words arranged along the axis. When viewed as a whole, the distribution of embeddings for each language exhibits a distinctive shape with spikes along axes. This spiky shape is universally observed across all languages.

We should discuss whether this spiky shape is real or not. ICA seeks axes that maximize non-Gaussianity (Hyvärinen and Oja, 2000). More generally, projection pursuit aims to find 'interesting' projections of high-dimensional data (Huber, 1985). However, these methods may detect apparent structures that are not statistically significant, particularly when $\gamma = d/n$ is large (Bickel et al., 2018). In the case of cross-lingual word embeddings, where $d = 300$ and $n = 50,000$, $\gamma = 0.006 \ll 1$ is very small. Therefore, it can be said that the chances of detecting apparent non-Gaussian structures are quite low. Taking into account the discovery of numerous common axes across all languages, it can be argued that the universal shape in the cross-lingual embedding distributions is real.

**The signature of ICA-transformed embeddings.** To investigate the characteristics of ICA-transformed word embeddings for each language, we plotted two measures of non-Gaussianity, namely skewness and kurtosis, along each axis in Fig. 16. Summary statistics for the skewness and kurtosis of the embeddings are given in Table 7. These results allow us to discern deviations from the isotropy of the distributions of word embeddings. All the kurtosis values for all axes in all languages were positive, indicating that the distributions are spreading more than the normal distribution. Although a general trend is observed in the plots, there are differences depending on the language. In particular, English shows the highest non-Gaussianity, indicating a most spiky shape. In contrast, Chinese and Hindi have the lowest non-Gaussianity and smoother shapes. It remains unclear whether these differences are language-specific or induced by the embedding training process.

# D   Details of experiment in Section 5

## D.1   Contextualized word embeddings

**Dataset.** We used `bert-base-uncased`, a pre-trained BERT model from the huggingface transformers library. This model was pre-trained on the BookCorpus (Zhu et al., 2015) and English Wikipedia. Sentences from the One Billion Word Benchmark (Chelba et al., 2014) were sequentially

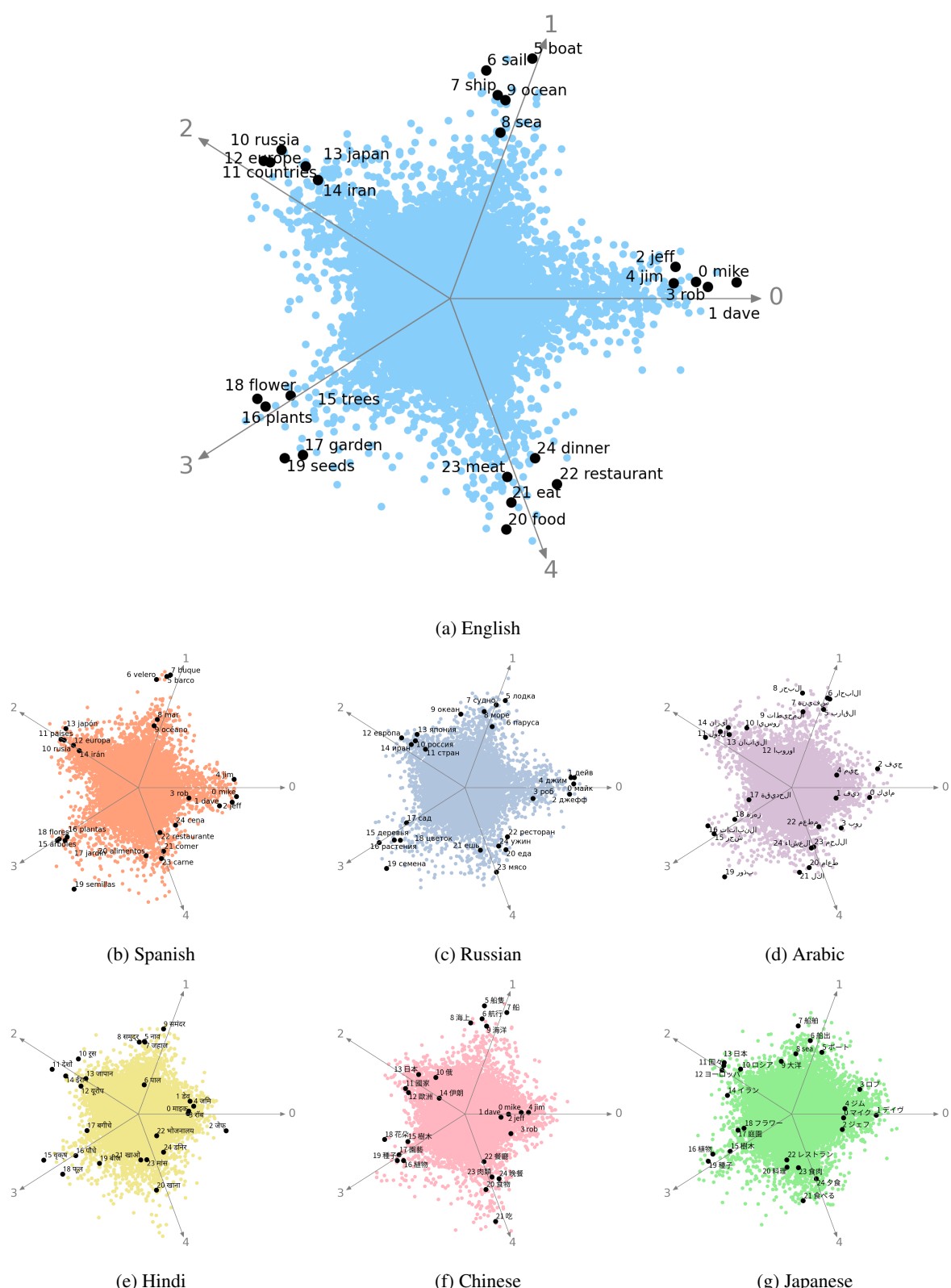

Figure 15: Normalized ICA-transformed word embeddings for seven languages. Scatterplots along the five axes presented in Fig. 2a were drawn by projecting the embeddings into two dimensions. For each language, all words in the vocabulary were plotted in respective colors. The 25 words in the heatmap were labeled and indicated by black dots.

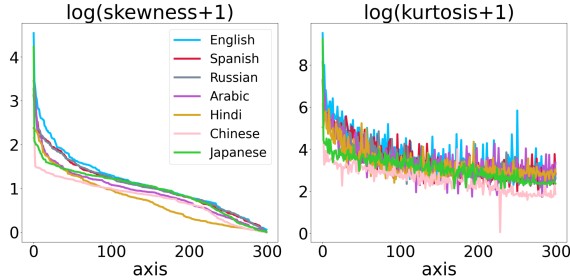

Figure 16: Skewness and kurtosis calculated for each axis of ICA-transformed word embeddings for seven languages. For each language, the axes are sorted in descending order of skewness.

| lang | skewness | | kurtosis | |
| --- | --- | --- | --- | --- |
| | mean | median | mean | median |
| EN | 3.25 | 1.84 | 129.63 | 30.12 |
| ES | 2.50 | 1.84 | 54.43 | 24.18 |
| RU | 2.49 | 1.75 | 48.65 | 22.63 |
| AR | 2.06 | 1.50 | 57.03 | 23.58 |
| HI | 1.53 | 1.01 | 35.70 | 20.08 |
| ZH | 1.38 | 1.34 | 15.62 | 11.84 |
| JA | 2.13 | 1.88 | 58.38 | 21.31 |

Table 7: Summary statistics of the skewness and kurtosis calculated for each axis of ICA-transformed word embeddings shown in Fig. 16.

inputted into BERT, generating 100,000 tokens, including [CLS] and [SEP]. Unlike static word embeddings like those from fastText, BERT yields dynamic word embeddings. Consequently, the same word can have different embeddings, labeled sequentially in their order of appearance, for instance, as *sea_0, sea_1*, and so on.

**PCA and ICA-transformed embeddings.** Similar to the cross-lingual experiments, we computed PCA-transformed and ICA-transformed embeddings for the obtained BERT embeddings. All the axes of the transformed embeddings were adjusted to have positive skewness.

**Alignment of axes via permutation.** We utilized the same English fastText embeddings used in Appendix C along with the BERT embeddings. We paired the fastText words with the BERT-labeled tokens. If the same word appeared $k$ times in the BERT tokens, we created $k$ pairs with that word. To adjust for this effect, the correlation coefficients were computed using the inverse frequency weight, $1/k$. Given that the dimensionality of fastText embeddings is 300 while that of BERT is 768, we greedily matched pairs of axes based on the most

significant correlation coefficients, thereby selecting 300 pairs of axes.

**Reordering axes.** Subsequently, we reordered the axes in descending order of the correlation coefficients between the aligned axes of fastText and BERT. The cross-correlation coefficients between the first 100 axes are presented in the middle panels of Fig. 3. ICA gives a strong alignment of the axes, while PCA gives a less favorable alignment.

**Word selection for visualization.** We normalized each embedding to have a norm of 1. We limited BERT tokens to those included in the fastText vocabulary. We selected the top 5 words from the fastText axis based on their component values. To examine the diversity of words, duplicates in singular and plural forms of nouns were disregarded. To verify the variations in dynamic word embeddings, we randomly selected 3 BERT embeddings for each word, instead of selecting those with the three largest component values. In the heatmaps, these three BERT embeddings were placed in descending order based on the component values. We performed this process for the initial 100 axes of both the ICA-transformed and PCA-transformed embeddings, and we presented the heatmaps of the components of the embeddings when selecting 500 words from fastText and 1,500 tokens from BERT.

**Analyzing the heatmaps.** In Fig. 7a, which illustrates the ICA-transformed embeddings, we observe similar patterns to the cross-lingual experiment. In the upper panels, representing 500 words and 1,500 tokens, the components of the top 5 words and sampled 15 tokens for each axis exhibit large values. This is due to the sparsity of the remaining components, resulting in significant values along the diagonal. Moreover, in the lower panels, we can observe consistent component patterns across most pairs of axes for the 25 words and 75 tokens.

On axis-1 of fastText embeddings, words such as *people* and *those* are relatively ambiguous and context-dependent. Considering that even in static fastText, their component values are smaller compared to other axes, it might be more challenging for a single axis to exhibit more significant components in dynamic embeddings of BERT.

On axis-2 of BERT embeddings, the component value for *shore_0* is nearly zero, while the component values for *shore_1* and *shore_2* are large. Upon reviewing the sentences containing

these words, *shore_0* appeared in the verb phrase *shore up*, while *shore_1* and *shore_2* were used to refer to *the land along the edge of a sea*. Therefore, axis-2 effectively represents a consistent meaning related to *ships-and-sea*.

Conversely, in Fig. 7b, which illustrates PCA-transformed embeddings, we did not observe the characteristics seen in Fig. 7a, mirroring the findings from the cross-lingual scenario.

### D.2 Image embeddings

**Dataset.** Beyond examining the universality across different languages and distinct word embedding models, we also investigated the universality across different image models. For the 1000 classes in ImageNet (Russakovsky et al., 2015), we assembled a dataset comprising 100,000 images, randomly selecting 100 images per class. As for the image models, we chose ViT-Base (Dosovitskiy et al., 2021) which is the backbone of CLIP (Radford et al., 2021), ResMLP (Touvron et al., 2023), Swin Transformer (Liu et al., 2021), ResNet (He et al., 2016), and RegNet (Han et al., 2018). The feature embeddings of these image models were extracted from their penultimate layer. We utilized the pre-trained weights from the huggingface PyTorch Image Models (Wightman, 2019) for these investigations, and Table 1 outlines the model types, weight types, and the dimensionalities of the embeddings.

**PCA and ICA-transformed embeddings.** Just as we did for fastText and BERT word embeddings, we computed PCA-transformed and ICA-transformed embeddings for image embeddings. All the axes of the transformed embeddings were adjusted to have positive skewness.

**Alignment of axes via permutation.** Similar to the cross-lingual experiment, where English fastText served as the reference model, we used ViT-Base as the reference model among the image models. In the process of computing the correlation coefficients between ViT-Base and another image model, the embeddings for the same image were treated as a pair. To align the axes of the ViT-Base embeddings with those of the other four image models, we employed the greedy matching approach based on the cross-correlation coefficients. Unlike the cross-lingual scenario, the image model embeddings have different dimensions. Therefore, we extracted only the axes from the

ViT-Base model that matched all the other models. As a result, there were 292 axes for the PCA-transformed embeddings and 276 axes for the ICA-transformed embeddings that were common among all the models.

**Diagnosing the axis alignment.** As an example, we presented the cross-correlation coefficients between the first 100 axes of the ViT-Base and ResMLP-12 models for both the PCA-transformed and ICA-transformed embeddings in the bottom panels of Fig. 3. As observed in previous experiments, the alignment between the ViT-Base and ResMLP-12 models is evident for the ICA-transformed embeddings. However, for the PCA-transformed embeddings, the alignment appears to be less clear and more ambiguous.

**Alignment with fastText.** Next, we considered the correspondence between the axes of ViT-Base embeddings and English fastText embeddings. It is important to note that the class names in ImageNet are not individual words but sentences, such as *'king snake, kingsnake'*. We parsed the class names into separate words and searched for those words in the vocabulary of English fastText, such as *'king'* and *'snake'*. For each ImageNet class, we randomly sampled 100 images, resulting in 100 pairs for *'king'* and 100 pairs for *'snake'* in the case of *'king snake, kingsnake'*. If a class name did not contain any of the words present in the vocabulary, that particular class was excluded from further analysis. When calculating the cross-correlation coefficients between the axes of ViT-Base embeddings and English fastText embeddings, each image-word pair was weighted inversely proportional to its frequency. Utilizing these cross-correlation coefficients, we employed the greedy matching approach to align the axes of ViT embeddings with the axes of fastText embeddings.

**Reordering axes.** Lastly, we rearranged the aligned axes of ViT-Base and fastText embeddings to ensure the correlation coefficients of the aligned axes are in descending order. The axes of other image models, previously matched with ViT-Base axes, were also rearranged according to the order of the ViT-Base axes.

**Analyzing the heatmaps.** We normalized each embedding to have a norm of 1. For each axis of ViT-Base, we selected the top 5 classes that have the highest average component values. For

each selected class, we randomly chose 3 images from the 100 images and sorted them in descending order based on the component value. The class name was parsed to find words in the English fastText vocabulary, and we selected the word with the largest component value on the corresponding axis. This process was applied to the first 100 axes for both PCA-transformed and ICA-transformed embeddings. The resulting heatmaps of the embeddings are displayed in Fig. 8.

From Fig. 8a for ICA-transformed embeddings, in the case of the 1,500 images and 500 words presented in the upper panels, the 15 images and 5 words on each axis exhibit significant values on that particular axis, much like in the cross-lingual and BERT experiments. This is because other components are sparse, causing large values to appear on the diagonal. The lower panels illustrate the heatmaps of the embeddings of 75 images and 25 words corresponding to the first five axes. For instance, axis-2 of ViT-Base is oriented toward alcohol-related concepts, the component values are large for images of bottles and beers. Models such as ResMLP-12, Swin-S, and RegNetY-200MF also manifest larger component values on axis-2 for images related to alcoholic beverages. When class names are partitioned into words, words selected on axis-2 of fastText include *beer, bottle*, and *wine*. The significant component values on the corresponding axes suggest a successful alignment between the axes of the image models and fastText.

Conversely, in Fig. 8b for PCA-transformed embeddings, the correspondence of axes is not evident. Furthermore, the interpretation of the top 5 classes associated with ViT-Base axes remains unclear.

It is important to note that ViT-Base, extracted from the pre-trained CLIP, aligns well with other models such as ResMLP-12 and Swin-S. This observation suggests that the decent alignment of ViT-Base with fastText is not merely a result of CLIP learning from both image and text.

## E   Details of experiment in Section 6

### E.1   Whitened embeddings

We introduce several whitening methods below. In Appendix A, we mentioned that they can be represented as

$$\mathbf{Y} = \mathbf{Z}\mathbf{R}$$

from the embeddings $\mathbf{Z}$ obtained through PCA and an orthogonal matrix $\mathbf{R}$. Although these whitened embeddings do not differ in terms of performance

on tasks based on inner products, they do differ in terms of interpretability, low-dimensionality, and cross-lingual performance. This study aims to identify the inherently interpretable subspace within pre-trained embeddings and analyze their sparsity and interpretability. Consequently, the baselines were chosen with this perspective in mind. In other words, embeddings learned directly from a corpus, which incorporate explicit sparsity and interpretability objectives in their optimization, are beyond the scope of this research.

**PCA.**   Let $\mathbf{\Sigma} = \mathbf{X}^\top\mathbf{X}/n$ be the covariance matrix of the row vectors of $\mathbf{X}$. In one implementation of PCA, eigendecomposition is performed as $\mathbf{\Sigma} = \mathbf{U}\mathbf{D}^2\mathbf{U}^\top$, where $\mathbf{U}$ is an orthogonal matrix consisting of the eigenvectors and $\mathbf{D}^2$ is a diagonal matrix consisting of the eigenvalues. Using these, the transformed matrix $\mathbf{Z}$ is computed as[8]:

$$\mathbf{Z} = \mathbf{X}\mathbf{U}\mathbf{D}^{-1}.$$

Another implementation directly computes $\mathbf{Z}$ from the singular value decomposition $\mathbf{X} = \mathbf{Z}\mathbf{D}\mathbf{U}^\top$.

**ICA.**   As mentioned in Section 3.2, ICA is represented as $\mathbf{S} = \mathbf{Z}\mathbf{R}_{\text{ica}}$ with the orthogonal matrix $\mathbf{R}_{\text{ica}}$. Thus $\mathbf{S}$ is whitened.

**ZCA-Mahalanobis whitening.**   The whitening transformation that minimizes the total squared distance between $\mathbf{X}$ and $\mathbf{Y}$ is computed as:

$$\mathbf{Y}_{\text{zca}} = \mathbf{X}\mathbf{\Sigma}^{-1/2},$$

where $\mathbf{\Sigma}^{-1/2} := \mathbf{U}\mathbf{D}^{-1}\mathbf{U}^\top$ (Bell and Sejnowski, 1997; Kessy et al., 2018). This can be expressed[9] as $\mathbf{Y}_{\text{zca}} = \mathbf{Z}\mathbf{R}_{\text{zca}}$, where $\mathbf{R}_{\text{zca}} = \mathbf{U}^\top$. Since the columns of $\mathbf{U}$ represent the directions of principal components, $\mathbf{Y}_{\text{zca}}$ simply rescales the original $\mathbf{X}$ along these directions without introducing any rotation.

**Crawford-Ferguson rotation family.**   A family of measures for the parsimony of matrix $\mathbf{Y}$ is proposed by Crawford and Ferguson (1970) as $f_\kappa(\mathbf{Y}) = (1 - \kappa)\sum_{i=1}^n\sum_{j=1}^d\sum_{k\neq j}^d y_{ij}^2 y_{ik}^2 + \kappa\sum_{k=1}^d\sum_{i=1}^n\sum_{j\neq i}^n y_{ik}^2 y_{jk}^2$, where $0 \leq \kappa \leq 1$ is a parameter.

Although initially proposed for post-processing the factor-loading matrix in factor analysis (Craw-

---

[8]$\mathbf{Z} = \mathbf{X}\mathbf{A}$ with $\mathbf{A} = \mathbf{U}\mathbf{D}^{-1}$ in Section 3.1.
[9]By noting $\mathbf{X} = \mathbf{Z}\mathbf{D}\mathbf{U}^\top$, we have $\mathbf{Y}_{\text{zca}} = \mathbf{X}\mathbf{\Sigma}^{-1/2} = (\mathbf{Z}\mathbf{D}\mathbf{U}^\top)(\mathbf{U}\mathbf{D}^{-1}\mathbf{U}^\top) = \mathbf{Z}\mathbf{D}\mathbf{D}^{-1}\mathbf{U}^\top = \mathbf{Z}\mathbf{U}^\top$.

ford and Ferguson, 1970; Browne, 2001), this measure can be used to find an optimal $\mathbf{R}$ by minimizing $f_\kappa(\mathbf{ZR})$, and use $\mathbf{ZR}$. Different values of $\kappa$ correspond to different rotation methods, such as quartimax ($\kappa = 0$), varimax ($\kappa = 1/n$), parsimax ($\kappa = (d-1)/(n+d-2)$), or factor parsimony ($\kappa = 1$).

If $\mathbf{Z}$ is a whitened matrix, the resulting matrix $\mathbf{Y} = \mathbf{ZR}$ is almost the same regardless of the choice of $\kappa$. This is because the second term of $f_\kappa(\mathbf{Y})$ satisfies $\sum_{k=1}^{d}\sum_{i=1}^{n}\sum_{j\neq i}^{n} y_{ik}^2 y_{jk}^2 = dn^2 - \sum_{k=1}^{d}\sum_{i=1}^{n} y_{ik}^4 = dn^2(1 + O_p(n^{-1}))$ as the rotated matrix $\mathbf{Y}$ is also whitened, i.e., $\sum_{i=1}^{n} y_{ik}^2/n = 1$. Therefore, the second term is almost constant with respect to $\mathbf{Y}$, and the result of the minimization is not significantly influenced by the value of $\kappa$.

## E.2 Unwhitened embeddings

We introduced some embeddings obtained by rotating the centered embeddings $\mathbf{X}$ without rescaling.

**PCA.** The diagonal elements of $\mathbf{D}$ represent the standard deviations of $\mathbf{X}$ in the directions of the principal components. Simply rescaling $\mathbf{Z}$ by these standard deviations results in the same scaling as $\mathbf{X}$, yielding

$$\mathbf{X}_{\text{pca}} := \mathbf{ZD} = \mathbf{XU}.$$

**ICA.** Since $\mathbf{S} = \mathbf{ZR}_{\text{ica}}$, we define

$$\mathbf{X}_{\text{ica}} := \mathbf{X}_{\text{pca}}\mathbf{R}_{\text{ica}} = \mathbf{X}(\mathbf{UR}_{\text{ica}}).$$

It should be noted that columns of $\mathbf{S}$ are intended to be independent random variables, while this is not the case for columns of $\mathbf{X}_{\text{ica}}$.

**ZCA.** Given that $\mathbf{Y}_{\text{zca}} = \mathbf{ZU}^\top$, we define

$$\mathbf{X}_{\text{zca}} := \mathbf{X}_{\text{pca}}\mathbf{U}^\top = \mathbf{X},$$

which brings us back to the original $\mathbf{X}$. This explains that ZCA involves only scaling without rotation.

**Crawford-Ferguson rotation family.** We simply apply the optimization procedure to $\mathbf{X}$. Specifically, we find an optimal $\mathbf{R}$ by minimizing $f_\kappa(\mathbf{XR})$, and use $\mathbf{XR}$. For unwhitened matrix $\mathbf{X}$, the minimization of $f_\kappa(\mathbf{XR})$ depends on the value of $\kappa$.

| | Quartimax | Varimax | Parsimax | Parsimony |
|---|---|---|---|---|
| DistRatio | 1.26 | 1.26 | 1.26 | 1.26 |

Table 8: Consistency of word meaning in the word intrusion task with Crawford-Ferguson rotation family.

## E.3 Interpretability: word intrusion task

**Selection of the intruder word.** Our objective is to assess the interpretability of the word embeddings $\mathbf{Y} \in \mathbb{R}^{n\times d}$, where each row vector $\mathbf{y}_i \in \mathbb{R}^d$ corresponds to a word $w_i$. In order to select the $w_{\text{intruder}}(a)$ for the set of top $k$ words of each axis $a \in \{1,\dots,d\}$, denoted as $\text{top}_k(a)$, we randomly chose a word from a pool of words that satisfy both of the following criteria simultaneously: (i) the word ranks in the lower $50\%$ in terms of the component value on the axis $a$, and (ii) it ranks in the top $10\%$ in terms of the component value on some axis other than $a$.

**Evaluation metric.** We adopted the following metric proposed by Sun et al. (2016).

$$\text{DistRatio} = \frac{1}{d}\sum_{a=1}^{d} \frac{\text{InterDist}(a)}{\text{IntraDist}(a)}$$

$$\text{IntraDist}(a) = \sum_{\substack{w_i, w_j \in \text{top}_k(a) \\ w_i \neq w_j}} \frac{\text{dist}(w_i, w_j)}{k(k-1)}$$

$$\text{InterDist}(a) = \sum_{w_i \in \text{top}_k(a)} \frac{\text{dist}(w_i, w_{\text{intruder}}(a))}{k}$$

In this formula, we defined $\text{dist}(w_i, w_j) = \|\mathbf{y}_i - \mathbf{y}_j\|$. Here, $\text{IntraDist}(a)$ denotes the average distance between the top $k$ words, and $\text{InterDist}(a)$ represents the average distance between the top words and the intruder word. The score is higher when the intruder word is further away from the set $\text{top}_k(a)$. Therefore, this score serves as a quantitative measure of the ability to identify the intruder word, thus it is used as a measure of the consistency of the meaning of the top $k$ words and the interpretability of axes.

**Results for Crawford-Ferguson rotation family.** Table 8 shows the DistRatio for whitened embeddings with the four different choices of $\kappa$ value. As we have discussed in Appendix E.1, there is no significant difference between the four rotation methods. So, we presented the result for the well-known varimax rotation in Table 2 of Section 6.1.

| | Tasks | k = 1 | | | | k = 10 | | | | k = 100 | | | | k = 300 | | | |
|---|---|---|---|---|---|---|---|---|---|---|---|---|---|---|---|---|---|
| | | ZCA | PCA | Vari. | ICA | ZCA | PCA | Vari. | ICA | ZCA | PCA | Vari. | ICA | ZCA | PCA | Vari. | ICA |
| Analogy | capital-common-countries | 0.00 | 0.03 | 0.17 | 0.57 | 0.26 | 0.37 | 0.44 | 0.90 | 0.94 | 0.95 | 0.94 | 0.97 | 0.97 | 0.97 | 0.97 | 0.97 |
| | capital-world | 0.00 | 0.01 | 0.08 | 0.33 | 0.13 | 0.20 | 0.34 | 0.74 | 0.87 | 0.87 | 0.89 | 0.92 | 0.92 | 0.92 | 0.92 | 0.92 |
| | currency | 0.00 | 0.00 | 0.29 | 0.20 | 0.05 | 0.07 | 0.30 | 0.28 | 0.17 | 0.20 | 0.24 | 0.27 | 0.23 | 0.23 | 0.23 | 0.23 |
| | city-in-state | 0.00 | 0.01 | 0.00 | 0.14 | 0.14 | 0.13 | 0.08 | 0.33 | 0.63 | 0.67 | 0.62 | 0.66 | 0.72 | 0.72 | 0.72 | 0.72 |
| | family | 0.00 | 0.06 | 0.19 | 0.21 | 0.16 | 0.23 | 0.37 | 0.51 | 0.73 | 0.70 | 0.75 | 0.75 | 0.81 | 0.81 | 0.81 | 0.81 |
| | gram1-adjective-to-adverb | 0.01 | 0.00 | 0.00 | 0.00 | 0.01 | 0.01 | 0.00 | 0.02 | 0.11 | 0.09 | 0.11 | 0.09 | 0.13 | 0.13 | 0.13 | 0.13 |
| | gram2-opposite | 0.00 | 0.00 | 0.00 | 0.00 | 0.00 | 0.00 | 0.00 | 0.01 | 0.09 | 0.12 | 0.11 | 0.13 | 0.14 | 0.14 | 0.14 | 0.14 |
| | gram3-comparative | 0.00 | 0.00 | 0.00 | 0.21 | 0.06 | 0.02 | 0.04 | 0.33 | 0.33 | 0.30 | 0.34 | 0.38 | 0.42 | 0.42 | 0.42 | 0.42 |
| | gram4-superlative | 0.00 | 0.00 | 0.00 | 0.12 | 0.02 | 0.02 | 0.01 | 0.14 | 0.25 | 0.25 | 0.25 | 0.29 | 0.32 | 0.32 | 0.32 | 0.32 |
| | gram5-present-participle | 0.00 | 0.02 | 0.00 | 0.03 | 0.04 | 0.04 | 0.05 | 0.19 | 0.32 | 0.30 | 0.35 | 0.38 | 0.40 | 0.40 | 0.40 | 0.40 |
| | gram6-nationality-adjective | 0.01 | 0.05 | 0.18 | 0.38 | 0.34 | 0.47 | 0.52 | 0.90 | 0.98 | 0.98 | 0.98 | 0.99 | 0.99 | 0.99 | 0.99 | 0.99 |
| | gram7-past-tense | 0.00 | 0.01 | 0.00 | 0.06 | 0.02 | 0.03 | 0.09 | 0.17 | 0.31 | 0.31 | 0.34 | 0.34 | 0.40 | 0.40 | 0.40 | 0.40 |
| | gram8-plural | 0.00 | 0.00 | 0.09 | 0.24 | 0.19 | 0.15 | 0.30 | 0.54 | 0.68 | 0.72 | 0.69 | 0.74 | 0.78 | 0.78 | 0.78 | 0.78 |
| | gram9-plural-verbs | 0.00 | 0.03 | 0.00 | 0.05 | 0.01 | 0.03 | 0.04 | 0.15 | 0.28 | 0.30 | 0.31 | 0.35 | 0.39 | 0.39 | 0.39 | 0.39 |
| | Average | 0.00 | 0.02 | 0.07 | **0.18** | 0.10 | 0.13 | 0.18 | **0.37** | 0.48 | 0.48 | 0.49 | **0.52** | 0.54 | 0.54 | 0.54 | 0.54 |
| Similarity | MEN | 0.08 | 0.09 | 0.24 | 0.36 | 0.30 | 0.30 | 0.39 | 0.55 | 0.65 | 0.66 | 0.67 | 0.68 | 0.70 | 0.70 | 0.70 | 0.70 |
| | WS353 | 0.14 | 0.21 | 0.31 | 0.33 | 0.31 | 0.32 | 0.46 | 0.66 | 0.64 | 0.67 | 0.68 | 0.71 | 0.71 | 0.71 | 0.71 | 0.71 |
| | MTurk | 0.10 | 0.07 | 0.26 | 0.37 | 0.25 | 0.33 | 0.47 | 0.57 | 0.54 | 0.52 | 0.54 | 0.58 | 0.55 | 0.55 | 0.55 | 0.55 |
| | RW | -0.03 | 0.02 | 0.02 | 0.09 | 0.08 | 0.07 | 0.16 | 0.16 | 0.35 | 0.31 | 0.29 | 0.33 | 0.35 | 0.35 | 0.35 | 0.35 |
| | SimLex999 | 0.05 | 0.00 | -0.01 | 0.03 | 0.08 | 0.13 | 0.10 | 0.14 | 0.25 | 0.25 | 0.22 | 0.23 | 0.26 | 0.26 | 0.26 | 0.26 |
| | SimVerb3500 | -0.03 | -0.02 | 0.01 | 0.00 | 0.03 | 0.04 | 0.04 | 0.04 | 0.12 | 0.12 | 0.11 | 0.12 | 0.15 | 0.15 | 0.15 | 0.15 |
| | Average | 0.05 | 0.06 | 0.14 | **0.20** | 0.17 | 0.20 | 0.27 | **0.36** | 0.43 | 0.42 | 0.42 | **0.44** | 0.45 | 0.45 | 0.45 | 0.45 |

Table 9: The performance of *whitened* embeddings (Appendix E.1) with components of top $k$ absolute value was evaluated. The values represent the top-10 accuracy for analogy tasks and the Spearman rank correlation for word similarity tasks.

| | Tasks | k = 1 | | | | k = 10 | | | | k = 100 | | | | k = 300 | | | |
|---|---|---|---|---|---|---|---|---|---|---|---|---|---|---|---|---|---|
| | | Orig. | PCA | Parsi. | ICA | Orig. | PCA | Parsi. | ICA | Orig. | PCA | Quarti. | ICA | Orig. | PCA | Vari. | ICA |
| Analogy | capital-common-countries | 0.01 | 0.02 | 0.32 | 0.56 | 0.32 | 0.49 | 0.84 | 0.94 | 0.98 | 0.98 | 0.97 | 0.99 | 0.98 | 0.98 | 0.98 | 0.98 |
| | capital-world | 0.00 | 0.01 | 0.21 | 0.32 | 0.20 | 0.36 | 0.64 | 0.82 | 0.92 | 0.92 | 0.93 | 0.95 | 0.95 | 0.95 | 0.95 | 0.95 |
| | currency | 0.00 | 0.00 | 0.27 | 0.22 | 0.04 | 0.09 | 0.28 | 0.28 | 0.23 | 0.25 | 0.24 | 0.26 | 0.27 | 0.27 | 0.27 | 0.27 |
| | city-in-state | 0.00 | 0.00 | 0.08 | 0.13 | 0.16 | 0.19 | 0.21 | 0.37 | 0.67 | 0.74 | 0.69 | 0.73 | 0.76 | 0.76 | 0.76 | 0.76 |
| | family | 0.00 | 0.05 | 0.26 | 0.21 | 0.25 | 0.41 | 0.51 | 0.53 | 0.78 | 0.80 | 0.84 | 0.83 | 0.86 | 0.86 | 0.86 | 0.86 |
| | gram1-adjective-to-adverb | 0.01 | 0.00 | 0.00 | 0.00 | 0.01 | 0.04 | 0.03 | 0.05 | 0.19 | 0.20 | 0.19 | 0.15 | 0.22 | 0.22 | 0.22 | 0.22 |
| | gram2-opposite | 0.00 | 0.00 | 0.00 | 0.00 | 0.02 | 0.02 | 0.03 | 0.03 | 0.09 | 0.16 | 0.14 | 0.18 | 0.20 | 0.20 | 0.20 | 0.20 |
| | gram3-comparative | 0.01 | 0.00 | 0.02 | 0.23 | 0.09 | 0.07 | 0.12 | 0.40 | 0.44 | 0.47 | 0.48 | 0.53 | 0.56 | 0.56 | 0.56 | 0.56 |
| | gram4-superlative | 0.00 | 0.00 | 0.00 | 0.12 | 0.03 | 0.03 | 0.05 | 0.17 | 0.31 | 0.37 | 0.36 | 0.36 | 0.43 | 0.43 | 0.43 | 0.43 |
| | gram5-present-participle | 0.00 | 0.04 | 0.03 | 0.01 | 0.06 | 0.10 | 0.11 | 0.23 | 0.44 | 0.43 | 0.44 | 0.48 | 0.52 | 0.52 | 0.52 | 0.52 |
| | gram6-nationality-adjective | 0.01 | 0.01 | 0.21 | 0.37 | 0.37 | 0.49 | 0.71 | 0.89 | 0.99 | 0.99 | 0.99 | 0.99 | 0.99 | 0.99 | 0.99 | 0.99 |
| | gram7-past-tense | 0.00 | 0.07 | 0.04 | 0.06 | 0.05 | 0.11 | 0.12 | 0.26 | 0.44 | 0.44 | 0.42 | 0.44 | 0.52 | 0.52 | 0.52 | 0.52 |
| | gram8-plural | 0.02 | 0.00 | 0.13 | 0.23 | 0.21 | 0.19 | 0.32 | 0.55 | 0.74 | 0.80 | 0.78 | 0.80 | 0.84 | 0.84 | 0.84 | 0.84 |
| | gram9-plural-verbs | 0.00 | 0.02 | 0.00 | 0.02 | 0.03 | 0.13 | 0.05 | 0.23 | 0.39 | 0.48 | 0.46 | 0.44 | 0.52 | 0.52 | 0.52 | 0.52 |
| | Average | 0.00 | 0.02 | 0.11 | **0.18** | 0.13 | 0.19 | 0.29 | **0.41** | 0.54 | 0.57 | 0.57 | **0.58** | 0.62 | 0.62 | 0.62 | 0.62 |

Table 10: The performance of *unwhitened* embeddings (Appendix E.2) with components of top $k$ absolute value was evaluated. The values represent the top-10 accuracy for analogy tasks.

| | Tasks | k = 1 | | | | k = 10 | | | | k = 100 | | | | k = 300 | | | |
|---|---|---|---|---|---|---|---|---|---|---|---|---|---|---|---|---|---|
| | | Orig. | PCA | Vari. | ICA | Orig. | PCA | Quarti. | ICA | Orig. | PCA | Vari. | ICA | Orig. | PCA | Vari. | ICA |
| Similarity | MEN | 0.12 | 0.13 | 0.34 | 0.36 | 0.35 | 0.34 | 0.53 | 0.60 | 0.68 | 0.68 | 0.69 | 0.70 | 0.72 | 0.72 | 0.72 | 0.72 |
| | WS353 | 0.13 | 0.11 | 0.40 | 0.32 | 0.33 | 0.40 | 0.58 | 0.67 | 0.68 | 0.70 | 0.73 | 0.73 | 0.73 | 0.73 | 0.73 | 0.73 |
| | MTurk | 0.16 | 0.21 | 0.42 | 0.38 | 0.28 | 0.46 | 0.62 | 0.62 | 0.61 | 0.64 | 0.62 | 0.65 | 0.64 | 0.64 | 0.64 | 0.64 |
| | RW | 0.03 | 0.06 | 0.03 | 0.13 | 0.13 | 0.19 | 0.14 | 0.22 | 0.37 | 0.34 | 0.34 | 0.37 | 0.38 | 0.38 | 0.38 | 0.38 |
| | Simlex999 | 0.05 | 0.06 | -0.00 | 0.06 | 0.10 | 0.17 | 0.12 | 0.16 | 0.25 | 0.26 | 0.25 | 0.25 | 0.27 | 0.27 | 0.27 | 0.27 |
| | SimVerb3500 | -0.03 | 0.03 | -0.06 | 0.02 | 0.03 | 0.07 | 0.06 | 0.07 | 0.14 | 0.14 | 0.14 | 0.14 | 0.16 | 0.16 | 0.16 | 0.16 |
| | Average | 0.08 | 0.10 | 0.19 | **0.21** | 0.20 | 0.27 | 0.34 | **0.39** | 0.46 | 0.46 | 0.46 | **0.47** | 0.48 | 0.48 | 0.48 | 0.48 |

Table 11: The performance of *unwhitened* embeddings (Appendix E.2) with components of top $k$ absolute value was evaluated. The values represent the Spearman rank correlation for word similarity tasks.

### E.4 Low-dimensionality: analogy task & word similarity task

**Analogy task.** We used the Google analogy dataset (Mikolov et al., 2013a), which includes 14 types of word relations for the analogy task. Each task is composed of four words that follow the relation $w_1 : w_2 = w_3 : w_4$. Using $w_1$, $w_2$ and $w_3$, we calculated $w_3 + w_2 - w_1$ and identified the top

| | | $k=1$ | | | | $k=10$ | | | | $k=100$ | | | | $k=300$ | | | |
|---|---|---|---|---|---|---|---|---|---|---|---|---|---|---|---|---|---|
| | | Quarti. | Vari. | Parsi. | FacParsi. | Quarti. | Vari. | Parsi. | FacParsi. | Quarti. | Vari. | Parsi. | FacParsi. | Quarti. | Vari. | Parsi. | FacParsi. |
| Analogy | Average | 0.07 | 0.07 | 0.07 | 0.07 | 0.18 | 0.18 | 0.18 | 0.18 | 0.49 | 0.49 | 0.49 | 0.49 | 0.54 | 0.54 | 0.54 | 0.54 |
| Similarity | Average | 0.14 | 0.14 | 0.14 | 0.14 | 0.27 | 0.27 | 0.27 | 0.27 | 0.42 | 0.42 | 0.42 | 0.42 | 0.45 | 0.45 | 0.45 | 0.45 |

Table 12: The average performance of the four rotation methods for *whitened* embeddings. The values are averaged over 14 analogy tasks and six word similarity tasks.

| | | $k=1$ | | | | $k=10$ | | | | $k=100$ | | | | $k=300$ | | | |
|---|---|---|---|---|---|---|---|---|---|---|---|---|---|---|---|---|---|
| | | Quarti. | Vari. | Parsi. | FacParsi. | Quarti. | Vari. | Parsi. | FacParsi. | Quarti. | Vari. | Parsi. | FacParsi. | Quarti. | Vari. | Parsi. | FacParsi. |
| Analogy | Average | 0.04 | 0.03 | **0.11** | 0.03 | 0.24 | 0.22 | **0.29** | 0.20 | **0.57** | 0.56 | 0.56 | 0.55 | 0.62 | 0.62 | 0.62 | 0.62 |
| Similarity | Average | 0.18 | **0.19** | 0.15 | 0.12 | **0.34** | 0.33 | 0.32 | 0.28 | 0.46 | 0.46 | 0.46 | 0.46 | 0.48 | 0.48 | 0.48 | 0.48 |

Table 13: The average performance of the four rotation methods for *unwhitened* embeddings. The values are averaged over 14 analogy tasks and six word similarity tasks.

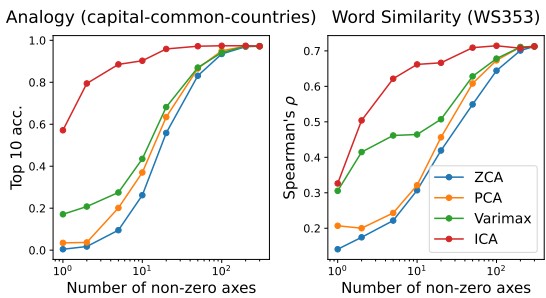

Figure 17: The performance of several whitened word embeddings when reducing the non-zero components. The top-10 accuracy for the analogy task (capital-common-countries) and the Spearman rank correlation for the word similarity task (WS353).

10 words with the highest cosine similarity to see if $w_4$ is included in them.

**Word similarity task.** We used MEN (Bruni et al., 2014), WS353 (Finkelstein et al., 2002), MTurk (Radinsky et al., 2011), RW (Luong et al., 2013), SimLex999 (Hill et al., 2015), and SimVerb-3500 (Gerz et al., 2016). They provide word pairs along with human-rated similarity scores. As the evaluation metric, we used the Spearman rank correlation coefficient between the human ratings and the cosine similarity of the word embeddings.

**Reducing the non-zero components.** For each transformed embedding $\mathbf{y} = (y_1, \ldots, y_d)$ and a specified value of $k$, we retained only the $k$ most significant components of $\mathbf{y}$. For example, if the components are $|y_1| \geq |y_2| \geq \cdots \geq |y_d|$, then we used $(y_1, y_2, \ldots, y_k, 0, \ldots, 0) \in \mathbb{R}^d$, where the $d-k$ least significant components are replaced by zero.

**Results.** The detailed results of the experiments presented in Section 6.2 are shown in Table 9 for

whitened embeddings and Tables 10, 11 for unwhitened embeddings. These results are derived from varying the number of non-zero components $k$, set to $k = 1, 10, 100,$ and 300 for each embedding. The performance of a specific case in analogy and word similarity tasks is also illustrated in Fig. 17.

For $k = 300$, all the components of the 300-dimensional word vectors were used as they are. Note that the performance for $k = 300$ is identical for all the whitened (or unwhitened) embeddings, because the difference is only their rotations, and both analogy tasks and similarity tasks are based on the inner product of embeddings.

Although there is a tendency for accuracy to decrease when reducing the number of non-zero components, it can be confirmed that the degree of decrease is smaller when using ICA compared to the other methods. The specific tasks depicted in Fig. 17 are those that achieved the highest performance at $k = 300$ in Table 9; capital-common-countries has the highest top-10 accuracy 0.97 in the analogy tasks, and WS353 has the highest Spearman correlation 0.71 in the word similarity tasks.

Results with embeddings that are transformed by the Crawford-Ferguson rotation family are shown in Table 12 for whitened embeddings and Table 13 for unwhitened embeddings. For whitened embeddings, there is no significant difference between the four rotation methods as discussed in Appendix E.1. So, we presented the results for the well-known varimax rotation in Section 6.2 and Table 9. For unwhitened embeddings, quartimax, varimax, and parsimax were similarly good. This result is consistent with the findings of Park et al. (2017). The best rotation was identified as boldface in Table 13

for the analogy task and word similarity task at each $k$, and the selected rotation method was used in Tables 10, 11.

### E.5 Cross-lingual alignment

| Dataset | | ES | FR | DE | IT | RU |
|---|---|---|---|---|---|---|
| | Pairs | 11965 | 10861 | 14669 | 9648 | 10883 |
| 157langs Train | Source | 4991 | 4994 | 4995 | 4993 | 4996 |
| | Target | 10154 | 9194 | 11221 | 8622 | 9512 |
| | Pairs | 2975 | 2943 | 3660 | 2585 | 2447 |
| 157langs Test | Source | 1500 | 1500 | 1500 | 1500 | 1500 |
| | Target | 2869 | 2855 | 3429 | 2532 | 2374 |
| | Pairs | 11977 | 10872 | 14677 | 9657 | 10887 |
| MUSE Train | Source | 5000 | 5000 | 5000 | 5000 | 5000 |
| | Target | 10166 | 9205 | 11229 | 8631 | 9516 |
| | Pairs | 2975 | 2943 | 3660 | 2585 | 2447 |
| MUSE Test | Source | 1500 | 1500 | 1500 | 1500 | 1500 |
| | Target | 2869 | 2855 | 3429 | 2532 | 2374 |

Table 14: The number of translation pairs, unique source English words, and unique target words for each target language.

**Datasets and visual inspection.** In addition to the fastText by Grave et al. (2018) utilized in Section 4, we employed fastText by MUSE (Lample et al., 2018). We refer to these word embeddings as 157langs-fastText and MUSE-fastText, and chose some of the languages common to these two datasets. For the cross-lingual alignment task, English (EN) was designated as the source language, while Spanish (ES), French (FR), German (DE), Italian (IT), and Russian (RU) were specified as the target languages. Following the same procedure as in Appendix C, we limited the vocabulary size to 50,000 in each language. The embeddings for the six languages are visualized in Fig. 18, by applying the same procedure as in Fig. 2.

**Applying a random transformation.** Note that MUSE-fastText already has pre-aligned word embeddings across languages. To resolve any such relationship across languages, we applied a random transformation to embeddings. For each embedding matrix $\mathbf{X} \in \mathbb{R}^{n \times d}$, we generated a random matrix $\mathbf{Q} \in \mathbb{R}^{d \times d}$ to compute $\mathbf{XQ}$. The random matrix was generated independently as

$$\mathbf{Q} = \mathbf{MLN},$$

where all the elements of $\mathbf{M}, \mathbf{N} \in \mathbb{R}^{d \times d}$ and $\mathbf{L} = \mathrm{diag}(l_1, \ldots, l_d) \in \mathbb{R}^{d \times d}$ are distributed independently. Specifically, the elements are $M_{ij}, N_{ij} \sim$

$\mathcal{N}(0, 1/d)$, the normal distribution with mean 0 and variance $1/d$, and $l_i \sim \mathrm{Exp}(1)$, the exponential distribution with mean 1. The random matrices $\mathbf{M}$ and $\mathbf{N}$ primarily induce rotation because they are roughly orthogonal matrices, while $\mathbf{L}$ induces random scaling.

**Word translation pairs.** We established the correspondence between the embeddings of English and the embeddings of other languages in the 157langs-fastText and MUSE-fastText datasets. To accomplish this, we followed the procedure outlined in Appendix C, but now we applied it to both the train set and the test set of MUSE dictionaries. The results of applying this procedure to the five target languages are presented in Table 14, where the source language is English. The train pairs were used for training supervised baselines and also for computing cross-correlation coefficients. The test pairs were used for computing the top-1 accuracy.

**Supervised baselines.** Two supervised baselines were considered to learn a linear transformation from the source embedding $\mathbf{X}$ to the target embedding $\mathbf{Y}$. We rearranged the word embeddings so that each row of $\mathbf{X}$ and $\mathbf{Y}$ corresponds to a translation pair, i.e., the meaning of the $i$-th row $\mathbf{x}_i$ corresponds to that of $\mathbf{y}_i$. We then computed the optimal transformation matrix $\mathbf{W} \in \mathbb{R}^{d \times d}$ that solves the least squares (LS) problem (Mikolov et al., 2013b):

$$\min_{\mathbf{W} \in \mathbb{R}^{d \times d}} \|\mathbf{XW} - \mathbf{Y}\|_2^2.$$

In the optimization of the Procrustes (Proc) problem, the transformation matrix $\mathbf{W}$ was restricted to an orthogonal matrix. Although LS is more flexible, the performance of cross-lingual alignment can possibly be improved by Proc (Xing et al., 2015; Artetxe et al., 2016). In these supervised methods, the two embeddings $\mathbf{X}$ and $\mathbf{Y}$ underwent centering and normalization as preprocessing steps.

**Cross-lingual alignment methods.** We considered both supervised and unsupervised transformations for cross-lingual alignment from the source language to the target languages. In the supervised transformation methods, LS and Proc, we first trained the linear transformation using both the source and target embeddings.

In the unsupervised transformation methods, PCA and ICA, we first applied the transformation individually to each language and then permuted

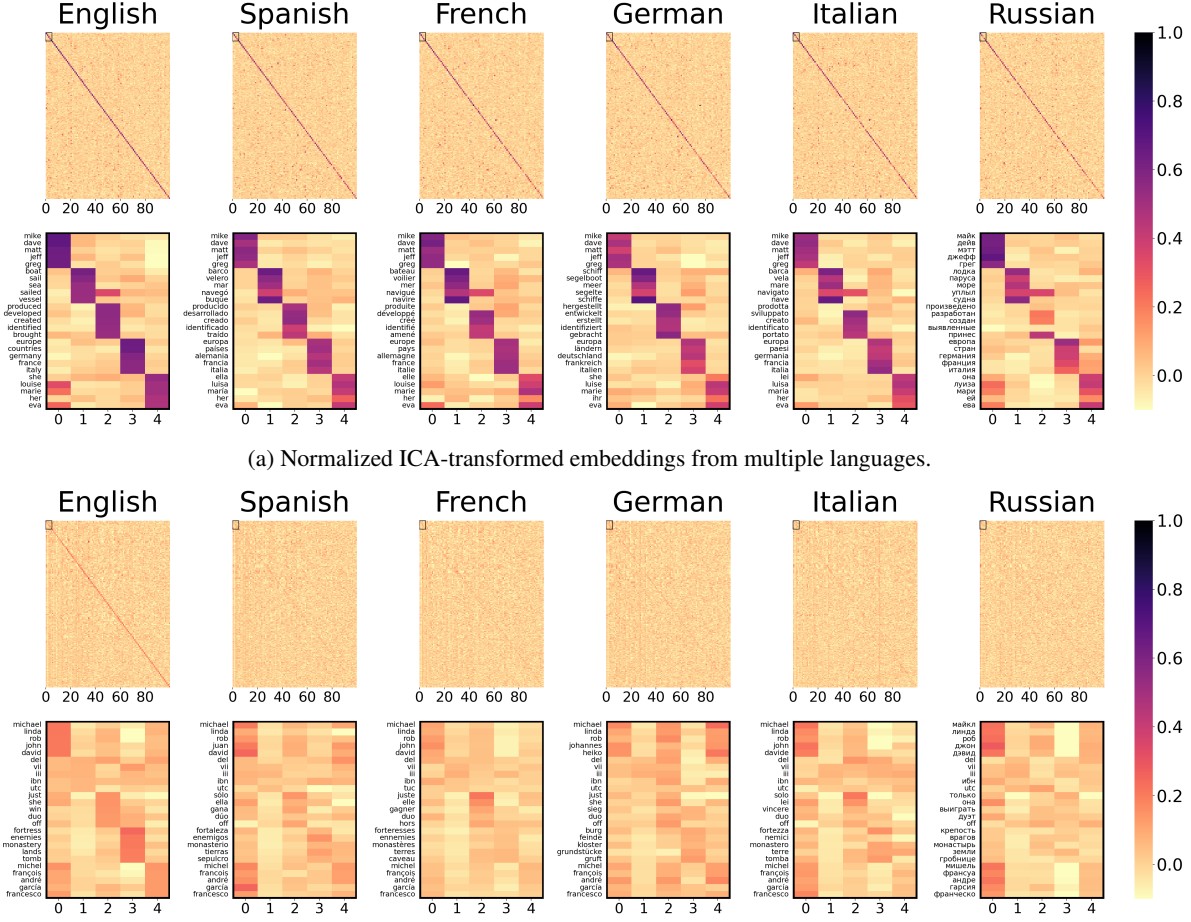

(a) Normalized ICA-transformed embeddings from multiple languages.

(b) Normalized PCA-transformed embeddings from multiple languages.

Figure 18: The embeddings for the six languages are visualized. They were obtained by applying the same procedure as in Fig. 2. Details of the datasets are presented in Appendix E.5.

| Dataset | Method | Original | | | | | | Random transformation | | | | | |
|---|---|---|---|---|---|---|---|---|---|---|---|---|---|
| | | ES | FR | DE | IT | RU | Avg. | ES | FR | DE | IT | RU | Avg. |
| 157langs-fastText | LS | 89.33 | 87.00 | 85.73 | 87.20 | 74.33 | 84.72 | 66.93 | 62.73 | 58.53 | 44.33 | 43.87 | 55.28 |
| | Proc | 88.40 | 86.93 | 86.80 | 86.80 | 75.80 | 84.95 | 26.00 | 20.93 | 20.00 | 14.73 | 11.93 | 18.72 |
| | ICA | 27.20 | 22.20 | 18.93 | 17.73 | 11.33 | 19.48 | 22.73 | 21.40 | 19.87 | 17.00 | 12.00 | 18.60 |
| | PCA | 1.53 | 0.87 | 0.80 | 0.80 | 0.60 | 0.92 | 0.80 | 0.67 | 0.67 | 0.60 | 0.47 | 0.64 |
| MUSE-fastText | LS | 84.47 | 83.87 | 80.53 | 80.53 | 65.13 | 78.91 | 60.60 | 59.73 | 53.53 | 44.33 | 39.00 | 51.44 |
| | Proc | 84.40 | 83.13 | 80.87 | 79.87 | 63.80 | 78.41 | 25.87 | 23.13 | 21.40 | 17.40 | 13.13 | 20.19 |
| | ICA | 53.00 | 54.27 | 45.80 | 43.60 | 19.67 | 43.27 | 52.93 | 52.80 | 45.93 | 43.33 | 21.53 | 43.31 |
| | PCA | 2.73 | 3.00 | 1.67 | 1.93 | 1.33 | 2.13 | 0.67 | 0.47 | 0.53 | 0.07 | 0.27 | 0.40 |

Table 15: The top-1 accuracy of the cross-lingual alignment task from English to other languages. Two datasets of fastText embeddings (157langs and MUSE) were evaluated with the two types of embeddings (Original and Random-transformation). LS and Proc are supervised transformations using both the source and target embeddings, while ICA and PCA are unsupervised transformations.

the axes based on the cross-correlation (see Appendix C). Although PCA and ICA are unsupervised transformations, the axis permutation is supervised because cross-correlation coefficients are computed from the embeddings of both languages.

**Evaluation metric.** For each word in the source language, we computed the transformed embedding and found the closest embedding from the target language in terms of cosine similarity. To mitigate the hubness problem, we used the CSLS

method (Lample et al., 2018) instead of the standard $k$-NN method. The top-1 accuracy was computed as the frequency of finding the correct translation word.

**Results.** The top-1 accuracy for all the five target languages is shown in Table 15, and only the average value is shown in Table 3. We used two datasets: 157langs-fastText and MUSE-fastText. Two types of embeddings were considered: one using the original word embeddings for all languages, and the other applying a random transformation to all the embeddings. The conclusions obtained in Section 6.3 regarding the average results of cross-lingual alignment hold true when considering each of the target languages.