# OpenReview forum: "Discovering Universal Geometry in Embeddings with ICA"
_EMNLP/2023/Conference — EMNLP 2023 Main_

### Official Review · Reviewer_QYwU · 2023-07-31

**Typos Grammar Style And Presentation Improvements:** NA
**Soundness:** 4

**Excitement:**

4: Strong: This paper deepens the understanding of some phenomenon or lowers the barriers to an existing research direction.

**Missing References:**

NA

**Paper Topic And Main Contributions:**

this paper found that words embedding can be expressed as a composition of a few intrinsic interpretable axes and that remain consistent across different languages, algorithms, and modalities.

**Questions For The Authors:**

1. The pretrained word embeddings are generally coming from complex nonlinear neural network. However, this work mainly use some linear analysis to get the results, like PCA and ICA. It is not well understood if those linear model are enough to analyze the nonlinear relationships among pretrained word embeddings.

**Reasons To Accept:**

1. This paper has some good findings that pretrained words embedding can be expressed as a composition of a few intrinsic interpretable axes and that remain consistent across different languages, algorithms, and modalities.

2. The paper is well written and provide many experimental results to validate their assumption. This findings may be useful for NLP model design.

**Reasons To Reject:**

1. The pretrained word embeddings are generally coming from complex nonlinear neural network. However, this work mainly use some linear analysis to get the results, like PCA and ICA. It is not well understood if those linear model are enough to analyze the nonlinear relationships among pretrained word embeddings.

2. It is better to provide the source code for  better reproducibility.

3. The technique contribution is limited as it mainly use PCA/ICA to do post-hot analysis.

**Reproducibility:**

3: Could reproduce the results with some difficulty. The settings of parameters are underspecified or subjectively determined; the training/evaluation data are not widely available.

**Reviewer Confidence:**

4: Quite sure. I tried to check the important points carefully. It's unlikely, though conceivable, that I missed something that should affect my ratings.

---

> ### Author Rebuttal · Authors · 2023-08-29
>
> Thank you for your comments and suggestions for improving the paper.
>
> ---
> # Question regarding the Linearity of Analytical Method
>
> > The pretrained word embeddings are generally coming from complex nonlinear neural network. However, this work mainly use some linear analysis to get the results, like PCA and ICA. It is not well understood if those linear model are enough to analyze the nonlinear relationships among pretrained word embeddings.
>
> Thank you for your insight.
>
> While we recognize the potential of nonlinear analyses, we think that even even linear analyses such as PCA and ICA can provide good analyses for understanding the nonlinear relationships between pretrained word embeddings. This is because  it's well known that linear relationships can provide good analyses for understanding the nonlinear relationships between pretrained word embeddings, e.g. **"king - man + woman = queen"**
>
> ---
>
> # Discussion on Limitation of Technical Contribution
>
> > The technique contribution is limited as it mainly use PCA/ICA to do post-hot analysis.
>
> Thank you for your insight. Indeed, the technical aspect of our research is limited to mainly using ICA to do post-hoc analysis. We agree that embeddings are so complicated that it requires more advanced technical contributions to deepen our understanding.
>
> On the other hand, despite the simplicity of our analysis, we have gained new insights into the geometry and interpretability of different types of embeddings. These results may be useful for future technical analyses of embeddings.
>
> Given the importance of the points you raise, we intend to include this perspective in the discussion of our camera-ready version.
>
> ---
>
> # Source Code Submission
>
> > It is better to provide the source code for better reproducibility.
>
> Thank you for your feedback. We recognize the importance of reproducibility. We plan to release the source code after making sure it's well organized. When we do, we will include a link to it in the paper.

---

### Official Review · Reviewer_fVPb · 2023-08-04

**Soundness:** 4

**Excitement:**

4: Strong: This paper deepens the understanding of some phenomenon or lowers the barriers to an existing research direction.

**Paper Topic And Main Contributions:**

This paper contributes an analysis of the geometry of embeddings for two domains: text and vision. The paper uses independent component analysis (ICA) to find a transformation of the embedding space in which the columns are independent components using the FastICA implementation in scikit-learn. The main claim put forth is that examples (e.g., words) can be represented as a set of intrinsic interpretable axes under ICA, and that the number of dimensions needed to represent each word is considerably less than the actual dimensions of the embeddings.

**Reasons To Accept:**

* Although there have been other studies that have applied ICA to word embeddings, this is the first to show that it yields embeddings that align to a significant degree across languages without explicit alignment between them (the "universal" claim).

* The study comprises an analysis of multiple text and image embedding models, including BERT, fastText, SGNS, ViT, ResMLT, Swin-S, ResNet, and RegNet.

* The findings regarding the degree of alignment of various independently trained embeddings under ICA, without explicit alignment steps, are somewhat surprising, and validate the "universal" claim.

**Reasons To Reject:**

* Given how much of the analysis hinges on ICA, it would have been helpful to provide more details in the *main text* about the FastICA approach. For example, does it make any assumptions? Is it exact and if not what sort of approximation error is incurred?

* Although ViT-base is extracted from CLIP which learns from both images and text (Appendix C), I think this paper may benefit from more focus on text. For example, what about popular contrastively trained text embeddings like sentence-bert (SBERT)? It would be surprising and interesting if the findings applied both to generatively trained and discriminatively trained representations. What about the impact of model size/capacity?

* It is somewhat well known that, across modalities, higher dimensionality embeddings often result in significantly better downstream performance. This is somewhat at odds with the conclusions of this paper that suggest that lower dimensional space is sufficient to capture the important information. What's lost in the ICA project? Are the embeddings less robust, for example?

**Reproducibility:**

4: Could mostly reproduce the results, but there may be some variation because of sample variance or minor variations in their interpretation of the protocol or method.

**Reviewer Confidence:**

4: Quite sure. I tried to check the important points carefully. It's unlikely, though conceivable, that I missed something that should affect my ratings.

---

> ### Author Rebuttal · Authors · 2023-08-29
>
> Thank you for your comments and suggestions for improving the paper.
> For the reasons to reject, below are our responses:
>
> ---
>
> # Question about the details of FastICA
>
> > Given how much of the analysis hinges on ICA, it would have been helpful to provide more details in the main text about the FastICA approach. For example, does it make any assumptions? Is it exact and if not what sort of approximation error is incurred?
>
> Thank you for pointing out the importance of providing more details about the FastICA approach, given its importance in our analysis. Changing the parameter settings of FastICA does not significantly affect the results. In the paper, we set the number of iterations for FastICA to 10,000. However, even when reduced to 200 iterations, important axes could still be extracted using ICA. In the revision, we will try to include some experimental results for other settings of ICA parameters.
>
> ---
>
> # Question about sentence embeddings
>
> > Although ViT-base is extracted from CLIP which learns from both images and text (Appendix C), I think this paper may benefit from more focus on text. For example, what about popular contrastively trained text embeddings like sentence-bert (SBERT)? It would be surprising and interesting if the findings applied both to generatively trained and discriminatively trained representations. What about the impact of model size/capacity?
>
> We appreciate your suggestion that our work could benefit from a more focused investigation of text embeddings. Your suggestion to consider embeddings such as Sentence-BERT (SBERT) is well taken. The potential impact of model size and capacity is indeed intriguing, and we look forward to exploring this aspect in future work.
>
> As a first step into sentence embeddings, we examined the ICA-transformed CLS token of BERT. This is because models such as Sentence-BERT and SimCSE, which are based on BERT, often use the CLS token as a representation for sentence embeddings that are further refined by fine-tuning. We observed several axes in the ICA-transformed BERT embeddings where CLS and SEP tokens were highly ranked. When we examined the original sentences associated with these special tokens, we found that they had consistent semantic meaning along certain axes. For example, we noticed different axes representing "headline" and "baseball" topics. Examples include
>
> The axis interpreted as "headline":
> - LOS ANGELES , California ( CNN ) -- A 24-year-old gang member was arrested Thursday in connection with...
> - PARIS ( Reuters ) - The body of the pilot of an Air France plane that crashed into the Atlantic on June...
> - NEW YORK ( AP ) - An undercover cop chased a Times Square scam artist through sidewalks crowded with...
>
> The axis interpreted as "baseball":
> - Hernandez escaped trouble in the third and fourth innings , then led off the fifth by hitting Byrd .
> - Victorino staked Philadelphia to a 1-0 lead in the first inning with a home run to right field His bloop...
> - The Angels took a lead in the third when rookie shortstop Jed Lowrie went to field a grounder with two...
>
> These results suggest that BERT's pre-training successfully encodes the semantics of a sentence into its special tokens. As Sentence-BERT and SimCSE undergo additional fine-tuning, it's possible that such trends in these models will become even more apparent. As we considered the axis matching of multiple image model embeddings from a single image in Fig. 5, we believe that an experiment could evaluate the axis matching of sentence embeddings derived from multiple models for a single sentence.
>
> ---
>
> # Quesiton about the embedding dimensionality
>
> > It is somewhat well known that, across modalities, higher dimensionality embeddings often result in significantly better downstream performance. This is somewhat at odds with the conclusions of this paper that suggest that lower dimensional space is sufficient to capture the important information. What's lost in the ICA project? Are the embeddings less robust, for example?
>
> As you pointed out, higher-dimensional embeddings tend to perform better in downstream tasks. ICA linearly transforms the embeddings to make them sparse, thus extracting a few essential axes for each word. Even when components with absolute values below a certain threshold are set to zero, the performance degradation is not significant (as shown in Fig. 6). However, these small-value components also encode some information, so setting them to zero results in a loss of information at some degree.

---

### Official Review · Reviewer_PoMJ · 2023-08-05

**Soundness:** 4

**Excitement:**

4: Strong: This paper deepens the understanding of some phenomenon or lowers the barriers to an existing research direction.

**Paper Topic And Main Contributions:**

The paper tries to understand the universal geometry in embeddings using independent principal analysis (ICA) and principle component analysis (PCA). The study shows that each embedding can be represented in low dimensions using ICA-transformed axes. In addition, different languages (English v.s. Spanish), algorithms (fasttext v.s. BERT), and modalities (words v.s. images) are considered in the study.

**Questions For The Authors:**

1. It's unclear to me how the five axes in Fig.1 (Left) are associated with the five words, dishes, cars, film, Italian, and Japanese.
2. It's not so clear what does it mean by "intrinsic interpretable axes" in the abstract.

**Reasons To Accept:**

1. The finding is interesting and strengthens the understanding of embeddings.
2. The result in Fig. 6 is encouraging that embeddings can be projected to a lower dimension using ICA while preserving the performance.

**Reasons To Reject:**

1. Clarity of the paper can be improved (see questions below).
2. The observation on the composition of semantic axes works the best for English fasttext but is not so clear for other languages and BERT.

===== After Rebuttal =====

Thanks to the author for the clear explanations in the response. All my questions are addressed. Thus, I decided to raise the Soundness to 4 (strong).

**Reproducibility:**

4: Could mostly reproduce the results, but there may be some variation because of sample variance or minor variations in their interpretation of the protocol or method.

**Reviewer Confidence:**

3: Pretty sure, but there's a chance I missed something. Although I have a good feel for this area in general, I did not carefully check the paper's details, e.g., the math, experimental design, or novelty.

---

> ### Author Rebuttal · Authors · 2023-08-29
>
> Thank you for your comments and suggestions for improving the paper.
> For the reasons to reject, below are our responses:
>
> ---
>
> # Question about Clarity of the Paper (1/2)
>
> > It's unclear to me how the five axes in Fig.1 (Left) are associated with the five words, dishes, cars, film, Italian, and Japanese.
>
> Thank you for your feedback. For clarity, we will add the following explanation.
>
> Words like dishes, meat, noodles, etc. have high values on the "axis 0", while words like cars, car, ferrari, etc. have high values on the "axis 1". In this way, each axis can be interpreted by a set of semantically related words. For the sake of interpretation of the axis, we have labeled the axes in the figure with the word with the highest value on each axis (e.g., 0:dishes, 1:cars, etc.). We are working on a clearer explanation and may update the figure and the caption in the camera-ready version. Thanks for your constructive feedback!
>
> ---
>
> # Question about Clarity of the Paper (2/2)
>
> > It's not so clear what does it mean by "intrinsic interpretable axes" in the abstract.
>
> We apologize for any confusion caused by our explanation. You're right; we didn't clarify this in the main text. Thank you for pointing this out. The axes of the ICA-transformed embeddings can be interpreted based on the set of words with high component values (interpretable axes). Since these axes are obtained by linearly transforming the original embeddings, they exist intrinsically in the original embeddings (intrinsic axes). We will explain this in the camera-ready version.
>
> ---
>
> # Question about the observation on the composition of semantic axes
>
> > The observation on the composition of semantic axes works the best for English fasttext but is not so clear for other languages and BERT.
>
> We apologize for the lack of clarity in our paper, and it appears that there may be a misunderstanding. Thank you for bringing this to our attention. We recognize the need to provide clearer explanations in our paper. In the section you point out, we illustrate how axis matching works well for English fastText as base embedding compared to other embeddings.
>
> English fastText is used as **the base embedding** (or a reference embedding) to be compared with other embeddings. In Fig. 2, we show the alignment of other languages with English, such as English-Spanish, ...,  English-Japanese. Similarly, in Fig. 4, we use English fastText as the base embedding to examine its alignment with BERT embedding.
>
> To further explain, for the "Normalized ICA-transformed word embeddings" languages closely related to English, such as Spanish, show significant component values for the top-ranked translated pairs (darker colors), while languages more distant to English, such as Chinese and Japanese, show smaller component values (lighter colors). Although fastText is a static embedding and BERT is dynamic, we observed good alignment between their axes and similar component values for the same words.

---

### Meta-Review · Area_Chair_cKW5 · 2023-09-11

**Recommendation:** 5

**Metareview:**

This paper undertakes an exploration of the universal geometry within embeddings through independent principal analysis (ICA). The study demonstrates that embeddings can be effectively represented in lower dimensions using ICA-transformed axes, encompassing different languages, algorithms, and modalities.

Overall, the reviewers generally find the paper sound. All reviewers agree that the paper is well-written and extensively supported by experimental results, enhancing its credibility and potential usefulness for NLP model design. In light of these merits, the consensus among reviewers is to accept the paper for publication.

---

### Decision · Program_Chairs · 2023-10-07

**Decision:**

Accept-Main

**Comment:**

This paper undertakes an exploration of the universal geometry within embeddings through independent principal analysis (ICA). The study demonstrates that embeddings can be effectively represented in lower dimensions using ICA-transformed axes, encompassing different languages, algorithms, and modalities.

Overall, the reviewers generally find the paper sound. All reviewers agree that the paper is well-written and extensively supported by experimental results, enhancing its credibility and potential usefulness for NLP model design. In light of these merits, the consensus among reviewers is to accept the paper for publication.